# BeST - A Novel Source Selection Metric for Transfer Learning

## Abstract

One of the most fundamental, and yet relatively less explored, goals in transfer learning is the efficient means of selecting top candidates from a large number of previously trained models (optimized for various "source" tasks) that would perform the best for a new "target" task with a limited amount of data. In this paper, we undertake this goal by developing a novel task-similarity metric (BeST) and an associated method that consistently performs well in identifying the most transferrable source(s) for a given task. In particular, our design employs an innovative quantization-level optimization procedure in the context of classification tasks that yields a measure of similarity between a source model and the given target data. The procedure uses a concept similar to *early stopping* (usually implemented to train deep neural networks (DNNs) to ensure generalization) to derive a function that approximates the transfer learning mapping without training. The advantage of our metric is that it can be quickly computed to identify the top candidate(s) for a given target task before a computationally intensive transfer operation (typically using DNNs) can be implemented between the selected source and the target task. As such, our metric can provide significant computational savings for transfer learning from a selection of a large number of possible source models. Through extensive experimental evaluations, we establish that our metric performs well over different datasets and varying numbers of data samples.

## 1 Introduction

Transfer Learning Pan and Yang (2010) Weiss et al. (2016) is a method to increase the efficacy of learning a target task by transferring the knowledge contained in a different but related source task. It is known that the effectiveness of supervised learning depends on the amount of labeled data. However, for various practical problems (e.g., medical imaging), collecting large quantities of labeled data might not be easy, as data collection and labeling is a tedious, expensive, and sometimes infeasible task (data is scarce, e.g., rare medical diseases). By employing transfer learning, we can enhance performance even with limited labeled data. Talking specifically of image classification, research in Oquab et al. (2014) showed how image representations learned with convolutional neural networks (CNNs) on large-scale annotated datasets can be efficiently transferred to other visual recognition tasks with limited data. The work in Yosinski et al. (2014) studied the impact of transferring features learned in different CNN layers and Long et al. (2015) describes how deeper layers can be more effectively transferred to a target CNN. The core idea of transfer learning is that different models trained for different sets of classes might learn some common features about the image in the initial layers that are not too task-specific. Recent studies show how transfer learning can be performed for CNNs by initializing the target neural network using feature-learned source CNN and adding a few dense layers to map the source model output to target labels.

From the literature Yosinski et al. (2014), it is evident that the choice of source model affects the target task performance as not every source shares similarities with the target, with some sources resulting in a phenomenon called *negative transfer* Wang et al. (2019). In transfer learning research the usual question is - *given a source and target task, how to transfer?* In contrast, in this work, we are trying to answer - *given a target task and many different source tasks, which source to choose for best transfer?* With the increasing availability of pre-trained learning models for a variety of classification tasks, it is now more important to assess the similarity between many existing source models $S_1, S_2, ...S_n$ and a new target task $T$ to find the best matching source task. The

straightforward approach for source selection is to train the target model using each source to find its accuracy. However, this is generally time-consuming given the complexity of large-scale neural networks. This calls for a quick task similarity metric that will help us rank a group of candidate source tasks without computationally intense neural network training for each of them. Specifically, this metric must measure the potential of a source model to be favorably utilizable by a given target task. We do not aim to propose the metric as an alternative to training but rather as a pre-processing step before training with the best source. Moreover, for it to be useful, we would like the metric to have the following properties –

- **Reliable in identifying good pairs:** The metric provides an accurate and reliable ranking for source models with high transfer learning performance (e.g., $> 90\%$ accuracy).
- **Time efficient:** Metric calculation should provide significant computational savings resulting in less time taken as compared to training a transfer learning model.
- **Architecture indifferent:** The metric does not use any knowledge of the architecture of the layers added on top of the pre-trained model. Hence, it should perform equally well if compared against different architectures if they have near-optimal accuracy performance.

In this work, we undertake this challenge for the scenario where source and target tasks are image classifiers. Our contributions can be summarized as — We propose a novel quantization-based approach to measure relatedness between a given source-target task pair to evaluate transfer learning performance. Our results show that our method can accurately rank source tasks for source models with a high transfer learning accuracy. We have a novel way to use the concept of generalization and early stopping, typically used in neural network training, to a problem outside the typical use case.

## 2  RELATED WORK

Quantization as a technique is not new to machine learning as it has been used to reduce the computational and memory costs of running inference by representing the weights and activations with low-precision data types like 8-bit integers instead of the usual 32-bit floating point. Several works Courbariaux et al. (2015); Gupta et al. (2015); Micikevicius et al. (2018), Gholami et al. (2021), talk about breakthroughs of half-precision and mixed-precision training. However, to the best of our knowledge, our approach to using quantization to transform model softmax output to evaluate task-transferability is novel. Authors in Dwivedi and Roig (2019) propose an approach to use Representation Similarity Analysis (RSA) to obtain a similarity score among tasks by computing correlations between models trained on different tasks. A study for automated source selection for transfer learning in CNNs using an entropy-based transferability measure is presented in Afridi et al. (2018). Methods like NCE Tran et al. (2019) and LEEP Nguyen et al. (2020) use the source and target labels to estimate transfer performance. In reality, we often do not have access to the source data but only to the source model as considered in our setup. Other methods like LogME You et al. (2021), GBC Pándy et al. (2022) and H-score Bao et al. (2019) use the source model embeddings and target data to estimate transferability. The work in Dai et al. (2019) evaluates transferability for Named Entity Recognition (NER) tasks in NLP. However, the method used domain-specific knowledge of NLP tasks, and hence cannot be generalized for other tasks (e.g., image classification). A similarity measure based on a restricted Boltzmann machine is proposed in Bou Ammar et al. (2014) to automatically select the best source task for transfer in the context of reinforcement learning.

## 3  SYSTEM MODEL

**Transfer learning architecture and training:** For many pre-trained models in real-world applications (e.g., ChatGPT, etc.), we have no access to the structures or parameters of the model. In such scenarios, we have to treat the source model as a *black-box*, i.e., we can only access the input and output of the source model. Therefore, we implement transfer learning as illustrated in Figure 1, where we append a custom model after the output of the source model to form the target model. The custom model is trained to minimize cross-entropy loss on training data $\mathcal{D}^{tr}$ using Adam optimizer. Early stopping is used to stop the training when the loss on validation data $\mathcal{D}^{val}$ no longer decreases.

**Models:** The source and target models, denoted by $f_s : \mathcal{X}^{\text{source}} \mapsto \mathcal{Y}^{\text{source}}$ and $f_t : \mathcal{X}^{\text{target}} \mapsto \mathcal{Y}^{\text{target}}$, are mappings from the source input set $\mathcal{X}^{\text{source}}$ to source output set $\mathcal{Y}^{\text{source}}$ and target input set

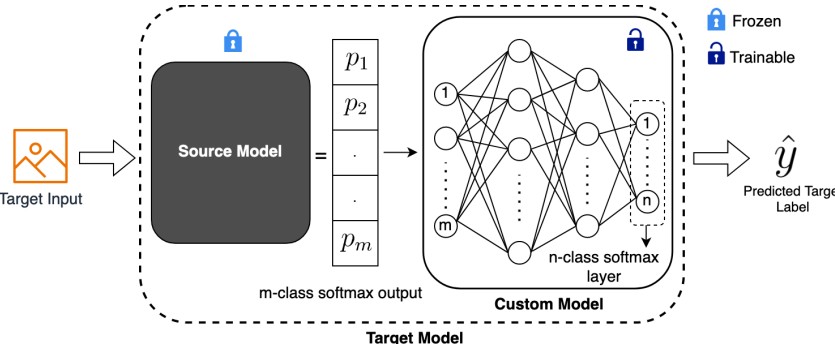

Figure 1: Transfer Learning architecture as concatenation of *black-box* source with a custom model.

$\mathcal{X}^{\text{target}}$ to target output set $\mathcal{Y}^{\text{target}}$ respectively. We assume that elements in both $\mathcal{X}^{\text{source}}$ and $\mathcal{X}^{\text{target}}$ are of the same dimension (e.g., images of the same size). This ensures that the target input can be directly fed to the source model without any pre-processing step. Unless stated otherwise, we assume the source and target models are $m$-ary and $n$-ary classifiers (generally $m \geq n$). The source output set is a set of softmax vectors and for a $m$-ary classifier, can be defined as $\mathcal{Y}^{\text{source}} = \{\mathbf{p} = (p_1, p_2, ..., p_m) | \sum_{i=1}^{m} p_i = 1, p_i \geq 0 \, \forall i\}$. The target output set is a set of labels and for a $n$-ary classification, $\mathcal{Y}^{\text{target}} = \{1, 2, ..., n\}$. The custom model, denoted by $f_c : \mathcal{Y}^{\text{source}} \mapsto \mathcal{Y}^{\text{target}}$, maps the source output set $\mathcal{Y}^{\text{source}}$ to the target output set $\mathcal{Y}^{\text{target}}$.

**Dataset:** We assume that we do not have access to the source data but only the target data denoted by $\mathcal{D} = \{\mathcal{D}^{tr}, \mathcal{D}^{val}, \mathcal{D}^{t}\}$, where $\mathcal{D}^{tr} = \{X^{tr}, Y^{tr}\}$, $\mathcal{D}^{val} = \{X^{val}, Y^{val}\}$, and $\mathcal{D}^{t} = \{X^{t}, Y^{t}\}$ represents the train, validation and test datasets. $X^{tr} = \{x_1^{tr}, ..., x_{n^{tr}}^{tr}\}$, $X^{val} = \{x_1^{val}, ..., x_{n^{val}}^{val}\}$, $X^{t} = \{x_1^{t}, ..., x_{n^{t}}^{t}\}$ denote the list of input data samples, while $Y^{tr} = \{y_1^{tr}, ..., y_{n^{tr}}^{tr}\}$, $Y^{val} = \{y_1^{val}, ..., y_{n^{val}}^{val}\}$, and $Y^{t} = \{y_1^{t}, ..., y_{n^{t}}^{t}\}$ denote the list of their respective labels. Here, $n^{tr}, n^{val}$, and $n^{t}$ denote the number of data samples for the three datasets respectively. We have assumed that the distribution of class labels for all $n$ classes is uniform[1] i.e. $\Pr(Y^{tr} = i) = 1/n, \forall i \in \{1, 2, ..., n\}$.

**Transferability:** The ability of a source model to enhance performance on a target task is referred to as transferability. It depends on the source model $f_s$, the custom model $f_c$, the target data $\mathcal{D}$, the optimizer parameters (parameters for Adam) and the training methodology. Given that we fixed the optimizer and the training methodology, transferability is represented as a function $T(f_s, f_c, \mathcal{D})$ and defined as the prediction accuracy of the trained target model on the unseen target test data $\mathcal{D}^t$.

## 4 OUR METHOD: BEST

**Problem Statement:** Given a set of $p$ pre-trained source models $S = \{f_s^1, f_s^2, ..., f_s^p\}$ and a target dataset $\mathcal{D}$, say $T_i = T(f_s^i, f_c, \mathcal{D})$ represents the ground truth of transferability for $i^{th}$ source. We want to define a transferability measure that takes the source model and target dataset as input, such that if $M_i$ represents the score given by the measure for $i^{th}$ source, the transferability ranks of the source models according to $\{M_i\}_{i=1}^{p}$ are very close to the ranks calculated using $\{T_i\}_{i=1}^{p}$.

To avoid using traditional neural network training to get the transferability of a source model $f_s$ to a target dataset $\mathcal{D}$, we need to derive an analytical function corresponding to the custom model $f_c$, that maximizes the prediction accuracy on target data. To ensure that its architecture-indifferent, it should only using the distribution of softmax output[2] and target labels. Given that we have limited target data samples, the estimation of this *continuous* distribution is imprecise.

To understand the need for quantization, assume that the source and target models are binary classifiers, and consider a '*hardmax*' case where the softmax output $[p_1, p_2]$ is transformed to a $2 \times 1$ one-hot discrete vector. In this case, there are only 4 choices for the custom model mapping (binary input to binary output) and it is easy to formulate the accuracy of each choice based on the estimation

---

[1]We do not want the custom model to be biased towards learning features for the input of a particular class.
[2]Produced when target input is fed to the source model.

of the *discrete* joint distribution. However, we lose the precision of information[3] in the transformation to one-hot vectors, affecting the accuracy of the mapping. The quantization approach helps us use the best of both worlds, where instead of mapping the softmax to a one-hot 'hard' vector, we can perform a custom transformation to a desired precision. It helps reduce our setup from a real-valued mapping (softmax to finite label) to a *quantized mapping* (one-hot vector to finite label) resulting in easier analytical analysis as it is a discrete function with a finite number of options.

Our algorithm (BeST) for best pre-trained source model selection calculates the optimal quantization level and corresponding metric for each model using target train and validation data. The source model with the highest metric is the model most suitable for the target task. The key part of this algorithm is how the metric is calculated and how well it can represent the performance of transfer learning. In Subsection 4.1, we define the quantization function for any $m$-ary source to $n$-ary target and Subsection 4.2 talks about the necessary notations and definitions. Subsection 4.3 talks about the variation of train and validation accuracies as the quantization level increases. Finally, in Subsection 4.4 we present the definition of our transferability metric and the algorithm to compute it.

## 4.1 NOVEL QUANTIZATION APPROACH

For an input $x \in \mathcal{X}^{\text{source}}$, the corresponding source softmax output $f_s(x) \in \mathcal{Y}^{\text{source}}$ represented as $\mathbf{p} = [p_1, p_2, ..., p_m]$, can be transformed to a one-hot vector with quantization level $q$ (or $q$-quantized for shorthand) using the quantization function $Q(\mathbf{p}, q) = \mathbf{p}^q$, where $\mathbf{p}^q$ is a $q^{(m-1)} \times 1$ vector and $p_i^q$ is 0 at every index $i$ except at index $i'$ given by Equation 1, where $p_{i'}^q = 1$.

$$i' = \begin{cases} \lfloor p_j q \rfloor q^{j-2} & ; \text{if } \exists j \in \{2, 3, ...m\} \text{ s.t. } p_j = 1 \\ \sum_{j=2}^{m} \lfloor p_j q \rfloor q^{j-2} \ + 1 & ; o.w. \end{cases} \tag{1}$$

We use the fact that we only need $(m-1)$ values to uniquely characterize a softmax vector with $m$ entries as the sum of values is 1. Figure 2 tries to explain the quantization process through an example where the source model is a ternary classifier and we transform it to a quantization level $q=3$. The vector $[p_2, p_3] = [0.7, 0.2]$ can be imagined to being mapped to a 2-D grid, where each dimension corresponds to representing one of $p_j$'s and is divided into 3 bins ($q=3$). The one-hot matrix can be unwrapped into a one-hot vector as in Figure 2. This can be extended to any $m \times 1$ softmax vector being mapped to a $(m-1)$-D grid and we can always obtain a one-hot representation.

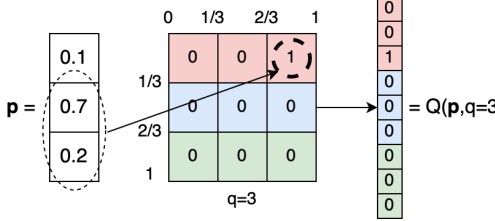

Figure 2: Quantization function explained through an example of a 3-class source model and $q=3$.

## 4.2 MATHEMATICAL FORMULATION

Let $\mathcal{H}^q = \{\mathbf{X} \in \{0,1\}^{q^{(m-1)}} : \sum_{i=1}^{q^{(m-1)}} X_i = 1\}$ denote the set of $q^{(m-1)} \times 1$ one-hot vectors. Consider set of all possible mappings $\mathcal{F}^q = \{\boldsymbol{\pi}^q | \boldsymbol{\pi}^q : \mathcal{H}^q \mapsto \mathcal{Y}^{\text{target}}\}$ from quantized source output set $\mathcal{H}^q$ to binary target label set $\mathcal{Y}^{\text{target}}$. Since $\mathcal{H}^q$ contains one-hot vectors, $\mathcal{F}_q$ can be equivalently defined as $\mathcal{F}_q = \{\boldsymbol{\pi}^q | \boldsymbol{\pi}^q : \mathbb{Z}^+_{\leq q^{(m-1)}} \mapsto \mathcal{Y}^{\text{target}}\}$, where $\boldsymbol{\pi}^q = [\pi_1^q, \pi_2^q, ..., \pi_{q^{(m-1)}}^q], \pi_i^q \in \mathbb{Z}^+_n$. Define $q$-quantized training and validation datasets denoted by $\mathcal{D}_q^{tr} = \{X_q^{tr}, Y^{tr}\}$ and $\mathcal{D}_q^{val} = \{X_q^{val}, Y^{val}\}$ respectively, where $X_q^{tr} = Q(f_s(X^{tr}), q)$ and $X_q^{val} = Q(f_s(X^{val}), q)$ represent the list of transformed input vectors for training and validation input data $X^{tr}$ and $X^{val}$. Let $(X \in \mathcal{X}^{\text{target}}, Y \in \mathcal{Y}^{\text{target}})$ denote random variables for the target input and label respectively. Assume that $P(q) = \Pr(X_q|Y)$ represents

---

[3][0.1, 0.9] and [0.4, 0.6] both have the same hardmax representation of [0,1].

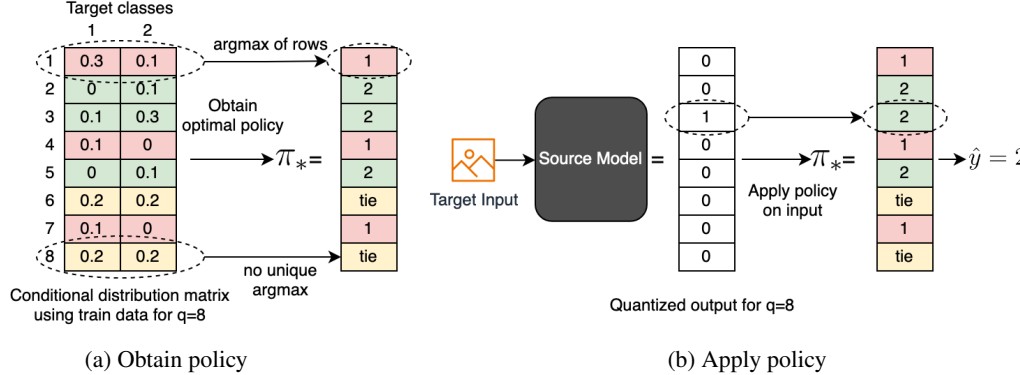

(a) Obtain policy             (b) Apply policy

Figure 3: Policy $\boldsymbol{\pi}_*^q$ explained through example with source and target models as binary classifiers.

the discrete conditional probability distribution of random variables representing $q$-quantized source output $X_q = Q(f_s(X), q)$ and target label $Y$. Since $X_q$ is a one-hot vector, we can equivalently define $P(q) = \Pr(\bar{X}_q | Y)$, where $\bar{X}_q = i$ if $X_{q,i} = 1$. Let $\hat{P}^{tr}(q) = \hat{\Pr}(\bar{X}_q | Y)$ represent empirical estimation of $P(q)$ using dataset $\mathcal{D}_q^{tr}$ and denote $\hat{P}_{i,j}^{tr}(q) = \hat{\Pr}(\bar{X}_q = i | Y = j)$, $i \in \mathbb{Z}_{q^{(m-1)}}^+, j \in \mathbb{Z}_n^+$.

We define the training accuracy $A^{tr}(\boldsymbol{\pi}^q)$ as the mapping accuracy of $\boldsymbol{\pi}^q$ on quantized training dataset $\mathcal{D}_q^{tr}$. It depends on the estimate of joint distribution $\hat{\Pr}(\bar{X}_q = i, Y = \pi_i^q)$ and using the uniform distribution assumption for samples per class, can be expressed as Equation 2.

$$A^{tr}(\boldsymbol{\pi}^q) = \sum_{i=1}^{q^{(m-1)}} \hat{\Pr}(\bar{X}_q^{tr} = i, Y^{tr} = \pi_i^q) = \frac{1}{n} \sum_{i=1}^{q^{(m-1)}} \hat{\Pr}(\bar{X}_q^{tr} = i | Y^{tr} = \pi_i^q) \qquad (2)$$

Let $\boldsymbol{\pi}_*^q$ represent the function that maximizes $A^{tr}(\boldsymbol{\pi}^q)$ for a fixed $q$. Hence, $A^{tr}(\boldsymbol{\pi}_*^q)$ and optimal function $\boldsymbol{\pi}_*^q$ are given in Equation 3 and 4 respectively.

$$A^{tr}(\boldsymbol{\pi}_*^q) = \max_{\boldsymbol{\pi}^q \in \mathcal{F}_q} \frac{1}{n} \sum_{i=1}^{q^{(m-1)}} \hat{\Pr}(\bar{X}_q^{tr} = i | Y^{tr} = \pi_i^q) = \frac{1}{n} \sum_{i=1}^{q^{(m-1)}} \max_{\pi_i^q} \hat{P}_{i,\pi_i^q}^{tr}(q) \qquad (3)$$

$$\pi_{*,i}^q = \begin{cases} \arg\max_j \hat{P}_{i,j}^{tr}(q) & \text{; if unique argmax exists} \\ r_k & \text{; if there are } k \text{ options for argmax} \end{cases} \qquad (4)$$

where $\pi_{*,i}^q = r_k$ means that there are $k$ options for *argmax* and the function $\pi_*^q$ would map to a uniform random choice between these $k$ classes. For each quantization level $q$, once we have the policy $\boldsymbol{\pi}_*^q$ that maximizes accuracy on $\mathcal{D}_q^{tr}$, its prediction accuracy on the validation dataset $\mathcal{D}_q^{val}$ denoted by $A^{val}(\boldsymbol{\pi}_*^q)$ can be empirically calculated as $A^{val}(\boldsymbol{\pi}_*^q) = (\sum_{i=1}^{n^{val}} \mathbb{1}(\pi_{*,j}^q = y_i^{val}))/n^{val}$, where $j = \arg\max x_{q,i}^{val}$. Figure 3 explains the process through an example considering the source and target models as binary classifiers i.e. $m=n=2$. Figure 3a explains obtaining $\boldsymbol{\pi}_*^q$ from the 2-D matrix representing the conditional probability $\hat{P}^{tr}(q)$ for $q = 8$. The entries in $\boldsymbol{\pi}_*^q$ are the *argmax* of the rows in the matrix and 'tie' represents no unique argmax. Figure 3b explains applying the policy $\boldsymbol{\pi}_*^q$ to target input where the predicted target label is the entry corresponding to the index in the one-hot quantized vector with 1. For 'tie', $\hat{y} = 0$ or 1 with equal probability.

## 4.3 QUANTIZATION TRADE-OFF

Referring to Figure 3a, imagine a *balls-in-bins* system, where the matrix represents the fraction of *balls* (quantized outputs) falling into 8 *bins* for two different kinds of *balls* (i.e. two target classes). Consider the first highlighted bin, where 10% of the data with target label 2 will be misclassified as label 1 resulting in reduced accuracy. Observe that there are a few rows with more 'overlap' than others (e.g., $[0.3, 0.1]$ has more overlap than $[0, 0.1]$), where more overlap means more samples are

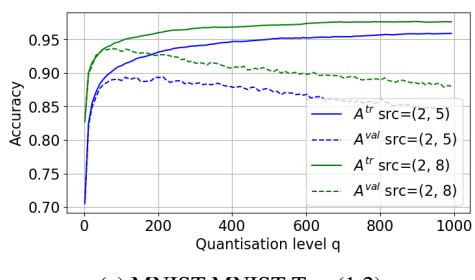 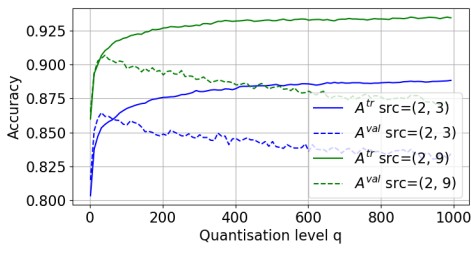

(a) MNIST-MNIST Tar=(1,2)

(b) CIFAR10-CIFAR10 Tar=(1,2)

Figure 4: Train-validation accuracy tradeoff where source and target tasks are binary classifiers. Tar=(1,2) and Src=(2,8) denote that the target and source tasks are to classify images of classes indexed 1 and 2, and 2 and 8 of the respective dataset (MNIST or CIFAR10).

misclassified. As $q$ increases, the softmax vectors are mapped to unique one-hot representations, resulting in the inputs, that earlier belonged to a the same bin, now belonging to different bins. This decreases the overlap in rows and resulting in increase in $A^{tr}(\boldsymbol{\pi}_*^q)$.

However, extremely large $q$ results in most of the bins receiving no balls at all. This means for the majority of the rows, the policy will predict a randomly chosen label out of the $n$ target labels resulting in poor validation accuracy. Figure 4a and 4b illustrate this variation of $A^{tr}(\boldsymbol{\pi}_*^q)$ and $A^{val}(\boldsymbol{\pi}_*^q)$ vs $q$ for MNIST-MNIST and CIFAR10-CIFAR10 setups, which mean the source and target tasks are classifiers trained on MNIST and CIFAR10 datasets respectively. Here as the quantization level increases, $A^{tr}(\boldsymbol{\pi}_*^q)$ increases but corresponding $A^{val}(\boldsymbol{\pi}_*^q)$ first increases and then starts decreasing.

**Theorem 4.1.** *Given that the source and target models are binary classifiers and the source softmax output is represented as a random vector $\boldsymbol{p} = [p_1, p_2]$, if true conditional probability distributions $f_1 = f_{(p_2|Y=1)}$ and $f_2 = f_{(p_2|Y=2)}$ are bounded, then as $q \to \infty$, $E[A^{val}(\boldsymbol{\pi}_*^q)] \xrightarrow{P} 1/2$.*

Theorem 4.1 says that if the underlying true conditional probability distributions of random variables $(p_2|Y = 1)$ and $(p_2|Y = 2)$ are bounded, then the expected validation performance on the given validation dataset $\mathcal{D}^{val}$ for policy that maximizes accuracy on target training data is as good as a coin flip policy (50% accuracy) as the quantization level goes to $\infty$ (proof in Appendix A). This result can be extended to a general $m$-ary source to binary target case as representation of $m$-ary softmax with quantization $q$ is mathematically equivalent to a binary softmax with quantization $q^{(m-1)}$.

### 4.4 Metric Definition and BeST Algorithm

Exploiting the trade-off explained in the previous section, we want to select a quantization level $q^*$ that maximizes the validation accuracy $A^{val}(\boldsymbol{\pi}_*^q)$ and this maximum accuracy, denoted as $A^{val}(\boldsymbol{\pi}_*^{q*})$ is defined as our metric. This approach to choose $q^*$ s.t. for $q > q^*$ the validation performance starts degrading, is analogous to *early stopping* in neural network training where we stop the training if the validation loss starts increasing. We need an upper bound on the search set for $q$ for practical implementation. Through simulations under various settings, we observed that $q^* \in (2, (n^{val}/n))$ as when $q > (n^{val}/n)$, the number of samples is way less than the number of rows ($q^{(m-1)}$) in the estimation of $P(q)$. Mathematically, $q^*$ and our metric $M$ are expressed in Equation 5.

$$q^* = \underset{q \in (2, (n^{val}/n)); \, q \in \mathbb{N}}{\text{argmax}} A^{val}(\boldsymbol{\pi}_*^q); \quad M = A^{val}(\boldsymbol{\pi}_*^{q*}) \tag{5}$$

Plots for different source-target pairs for different datasets in Figure 4 suggest that $A^{val}(\boldsymbol{\pi}_*^q)$ behaves approximately as an unimodal function (more in Appendix E). Hence, we can use *ternary-search* as a heuristic to find $q^*$ using Algorithm 1. In step 1, we ensure that there are an equal number of samples from each target class and step 2 initializes the left and right variables bounding the search space for $q^*$. Steps 3-16 uses ternary search to update the left and right pointers to calculate and compare $A^{val}(\boldsymbol{\pi}_*^q)$ to come closer to $q^*$. The search stops if the difference between the left and right pointer is within *tolerance* or if the maximum number of steps (*max_steps*) is reached. Finally, the metric is the average of $A^{val}$ values for the policy $\boldsymbol{\pi}_*^q$ at the final left and right quantization levels.

---

**Algorithm 1:** BeST: Quantisation based Task-Similarity Metric

---

**Input:** $f_s, \mathcal{D}^{tr} = \{X^{tr}, Y^{tr}\}, \mathcal{D}^{val} = \{X^{val}, Y^{val}\}$, tolerance, max_steps
**Output:** $M$

1 $\mathcal{D}^{tr}, \mathcal{D}^{val} \leftarrow$ modified $\mathcal{D}^{tr}, \mathcal{D}^{val}$ s.t. number of samples for each target class is equal;

2 $L \leftarrow 2, R \leftarrow n^{val}/n =$ num of samples per class in $\mathcal{D}^{val}$;

3 **while** $(|L - R| >$ tolerance$)$ and $($step $<$ max_steps$)$ **do**

4     $m_1 \leftarrow \lfloor L + (R - L)/3 \rfloor, \ m_2 \leftarrow \lfloor L - (R - L)/3 \rfloor$;

5     $(\mathcal{D}^{tr}_{m_1}, \mathcal{D}^{val}_{m_1})$ and $(\mathcal{D}^{tr}_{m_2}, \mathcal{D}^{val}_{m_2}) \leftarrow$ quantized version of datasets $(\mathcal{D}^{tr}, \mathcal{D}^{val})$ at $q = m_1, m_2$;

6     Calculate $\hat{P}^{tr}(m_1)$ and $\hat{P}^{tr}(m_2)$ using $\mathcal{D}^{tr}_{m_1}$ and $\mathcal{D}^{tr}_{m_2}$;

7     $\pi^{m_1}_* \leftarrow$ optimal policy using $\hat{P}^{tr}(m_1)$,    $\pi^{m_2}_* \leftarrow$ optimal policy using $\hat{P}^{tr}(m_2)$;

8     Calculate $A^{val}(\pi^{m_1}_*)$ and $A^{val}(\pi^{m_2}_*)$ using $\pi^{m_1}_*$ and $\pi^{m_1}_*$ to predict on $\mathcal{D}^{val}_{m_1}$ and $\mathcal{D}^{val}_{m_2}$;

9     $L_{val} \leftarrow A^{val}(\pi^{m_1}_*), \ R_{val} \leftarrow A^{val}(\pi^{m_2}_*)$;

10     **if** $L_{val} < R_{val}$ **then**

11        $L \leftarrow m_1$;

12     **else**

13        $R \leftarrow m_2$;

14     **end**

15     step $\leftarrow$ step $+ 1$;

16 **end**

17 $(\mathcal{D}^{tr}_L, \mathcal{D}^{val}_L)$ and $(\mathcal{D}^{tr}_R, \mathcal{D}^{val}_R) \leftarrow$ quantized version of datasets $(\mathcal{D}^{tr}, \mathcal{D}^{val})$ at $q = L, R$;

18 Calculate $\hat{P}^{tr}(L)$ and $\hat{P}^{tr}(R)$ using $\mathcal{D}^{tr}_L$ and $\mathcal{D}^{tr}_R$;

19 $\pi^L_* \leftarrow$ optimal function using $\hat{P}^{tr}(L)$;    $\pi^R_* \leftarrow$ optimal function using $\hat{P}^{tr}(R)$;

20 Calculate $A^{val}(\pi^L_*)$ and $A^{val}(\pi^R_*)$ using $\pi^L_*$ and $\pi^R_*$ to predict on $\mathcal{D}^{val}_L$ and $\mathcal{D}^{val}_R$;

21 $M \leftarrow \dfrac{A^{val}(\pi^L_*) + A^{val}(\pi^R_*)}{2}$

---

## 5 Experiments

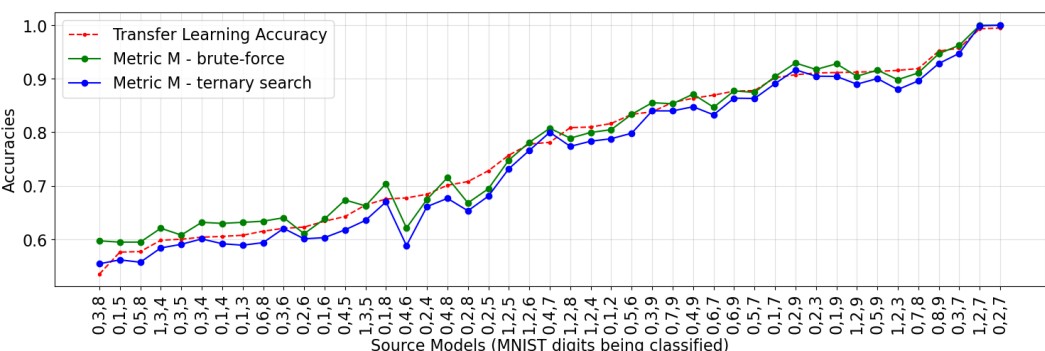

Figure 5: Comparison of ranks predicted by metric and ground truth for 3-class source to 2-class target transfer in MNIST-MNIST TL setup with $\sim 500$ data samples using 5-layer custom model.

**Datasets and experimental settings:** The experiments assess our metric's effectiveness in ranking source models with classifiers trained on two datasets – MNIST and CIFAR10. We consider 3 transfer learning setups (TL setups) – MNIST-MNIST, CIFAR10-CIFAR10 and CIFAR10-MNIST, where the first and second datasets correspond to the dataset source and target models are trained on. To emulate the limited data setup, we use *tl-frac* parameter where *tl-frac*=0.01 means choose a random subset with only 1% of the entire dataset (100 samples for 10000 sample dataset). Practically, *tl-frac*=0.01, 0.03, and 0.05 corresponds to around 50, 150, and 250 data samples per class with 80%-20% split for train and validation. In every transfer learning setup, we assess 45 unique source models for a specific target (details in Appendix C), ranking them across 20 iterations, using a different subset in each iteration to avoid data-specific bias in the evaluation. For our metric calcu-

lation using Algorithm 1, *tolerance* and *max-steps* parameters are set to 5 and 20 steps respectively. Tar/Src=$(i, j)$ is used as shorthand to denote the target (or source) model trained to classify images from classes indexed $i$ and $j$ (e.g., Tar=$(1, 2)$ in MNIST-MNIST refers to classifying digits 0 and 1).

**Model architectures and metric evaluations:** We train different source models for a given dataset using a fixed model architecture (Appendix B) to ensure a fair comparison. We consider a fully connected DNN for our custom model with 2-layer and 5-layer architectures (Appendix B), to show that our metric is architecture-indifferent. Early stopping stops the training if the validation error doesn't decrease by 0.01 in the next 20 epochs. We introduce an accuracy *threshold* parameter to select a subset of 45 source models to run our experiments for a given target task (e.g., *threshold*=0.9 means all source models with transfer learning accuracy > 90% are chosen). To avoid the true rankings being hypersensitive to small differences in transfer learning accuracy, rank $i$ assigned to a source model by our metric is counted correct if its transfer learning accuracy is within 3% of the transfer learning accuracy of the source with true rank $i$. We evaluate the ranking performance of our metric on the fraction of accurate rankings, the mean deviation of predicted ranks from true ranks, and the factor of time savings from training the custom model.

Figure 5 presents a comparison between the true ranks of various source models (red) with the predicted ranks (green and blue) in a transfer learning setup where source models are ternary classifiers and the target task is to classify images of digits 2 and 7. The metric values for the green curve use the brute-force method to find the optimal $q^*$ and ranks. Observe that the ranks according to ternary search and brute-force search are very close, supporting our unimodal approximation in Section 4.4. Observe that one of the best source models to transfer from is the one trained to classify the digits (0,2,7) and (1,2,7), which is intuitive as these already know how to classify digits 2 and 7.

## 5.1 BINARY CLASSIFIERS

We first evaluate the basic setup where the source and target are binary classifiers. Figure 6 shows the fraction of correct ranks assigned using our metric as compared to the true ranks, as *threshold* parameter changes for different TL setups using the 5-layer custom model. We can observe that the fraction of correct ranks increases with an increase in *threshold*, with the best performance for source models with transfer learning accuracy of > 90%. This means that our metric works better in ranking good transfer learning pairs.

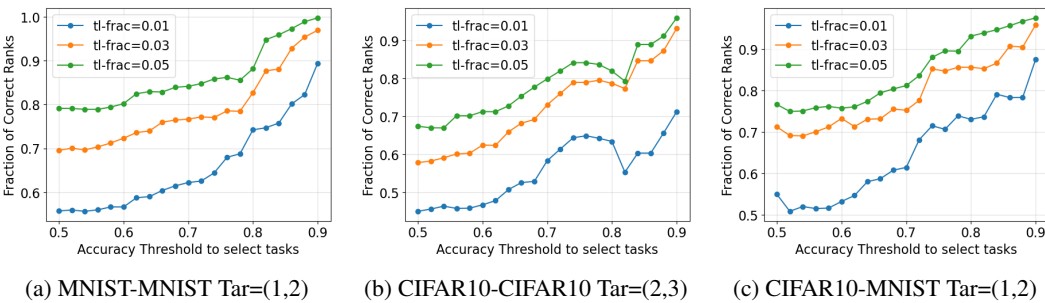

| (a) MNIST-MNIST Tar=(1,2) | (b) CIFAR10-CIFAR10 Tar=(2,3) | (c) CIFAR10-MNIST Tar=(1,2) |

Figure 6: Fraction of accurate ranks Vs *threshold* for different dataset sizes for 5-layer custom model.

Figure 7 shows the fraction of correct ranks for high *threshold* with different *tl-frac* values using 2 different target tasks for each TL setup for both custom models. Figures 7a, 7b, 7c suggest that our metric consistently performs well when ranking source models with > 80% transfer learning accuracy across all 3 TL setups and different target tasks. Observe that generally the performance of 5-layer custom model (maroon and dark green) is higher than that of 2-layer (red and green) since 5-layer has more capability to achieve the optimal generalization performance. However, the performance for both custom models are > 60% across TL setups indicating that the ranks assigned by the metric are not a random guess. Both Figures 6 and 7 suggest that the fraction of correct ranks increase with the datasize (*tl-frac*) across the 3 TL setups for different target tasks for each *threshold*.

Table 1 presents details on the mean and standard deviation of the deviation in ranks for MNIST-MNIST TL setup. We can observe that the mean deviation decreases with an increase in *tl-frac*

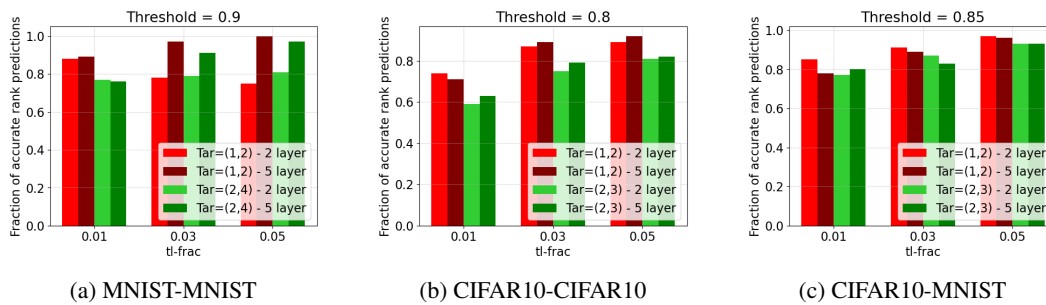

(a) MNIST-MNIST  (b) CIFAR10-CIFAR10  (c) CIFAR10-MNIST

Figure 7: Rank similarity performance of metric for source tasks with high TL accuracy for given target tasks for varying data sizes (given by *tl-frac*) using both 2-layer and 5-layer custom model.

i.e. data size, and also with an increase in *threshold*. This aligns with our earlier observation that our metric performs the best for source models with $> 90\%$ transfer learning accuracy. For *threshold*=0.9, we can particularly observe that even using just 100 samples (*tl-frac*=0.01) the predicted ranks are off by less than 2 ranks on average, which reduces even further to less than 1 rank when using 500 samples (*tl-frac*=0.05). The small standard deviation values suggest that there is less variability in the deviation of ranks establishing consistent performance across source-target pairs.

Table 1: Statistics for deviation of predicted ranks from true ranks for MNIST-MNIST TL Setup with target task to classify images of digits 1 and 3.

| Custom Model | Threshold | tl-frac=0.01 | | | tl-frac=0.03 | | | tl-frac=0.05 | | |
|---|---|---|---|---|---|---|---|---|---|---|
| | | 0.5 | 0.7 | 0.9 | 0.5 | 0.7 | 0.9 | 0.5 | 0.7 | 0.9 |
| 2-layer | Mean | 8.52 | 5.14 | **0.85** | 8.11 | 5.34 | **0.81** | 8.44 | 5.53 | **0.67** |
| | Std | 1.4 | 1.2 | 0.59 | 1.35 | 1.18 | 0.65 | 1.75 | 1.21 | 0.54 |
| 5-layer | Mean | 8.8 | 6.22 | **1.27** | 6.17 | 4.01 | **0.49** | 4.41 | 3.15 | **0.18** |
| | Std | 1.48 | 0.95 | 0.58 | 1.62 | 1.26 | 0.51 | 1.50 | 1.22 | 0.26 |

One of the most important aspects of evaluating our metric's performance is the computational savings offered compared to traditional training. A comparison of the average time taken to calculate our metric and the time taken to train a target model for a particular TL setup is presented in Table 2 (we used M3 MacBook Pro with 24 GB RAM). Here time taken is a measurement of CPU time given in seconds (not wall time). The CPU time measures the sum of the total time taken by all the CPU cores in use and is the true cost of computation as it is not affected by multi-core computation. Our metric provides significant time savings for each TL setup, with as high as 57 times faster computation. We observe that the computational benefits are reflected across data sizes indicating robustness. We also observe that the time taken by the metric scales sub-linearly with the data size (*tl-frac*) which suggests that it can work with a large number of samples.

Table 2: Improvement in time taken for metric calculation vs target neural network training for all 3 TL setups for 5-layer custom model for binary classifier case. Values given in CPU seconds.

| | MNIST-MNIST Tar=(2,4) | | | CIFAR10-CIFAR10 Tar=(1,2) | | | CIFAR10-MNIST Tar=(2,4) | | |
|---|---|---|---|---|---|---|---|---|---|
| tl-frac | NN | Metric | Eff. | NN | Metric | Eff. | NN | Metric | Eff. |
| 0.01 | 4.49 | 0.11 | ×**41** | 7.71 | 0.17 | ×**44** | 11.12 | 0.25 | ×**45** |
| 0.03 | 12.76 | 0.23 | ×**57** | 24.84 | 0.48 | ×**52** | 29.65 | 0.76 | ×**39** |
| 0.05 | 12.39 | 0.27 | ×**46** | 23.68 | 0.58 | ×**41** | 34.81 | 0.89 | ×**39** |

## 5.2 MULTI-CLASS CASE

We now consider the results for the multiclass case where we consider 3 settings – 3-class source to binary target, 4-class source to binary target, and 4-class source to 3-class target. Similar to Figure 7, we present the fraction of correct rank statistics in Figure 8 for the multiclass case for the 5-layer

custom model, where the x-axis represents the TL setup with a specific target task. To demonstrate that our metric works best when identifying the best-performing source-target pairs, we consider high *threshold* values (0.9 for Figure 8a and 8b and 0.85 for 8c). However, note that we omitted the CIFAR10-MNIST plot for Figure 8c since the maximum transfer learning accuracy for 4-class source to 3-class target was around $75\%$ (maybe need a larger custom model to achieve better). We can observe in each of Figures 8a, 8b, 8c, the fraction of correct ranks increases with an increase in *tl-frac* (light to dark bar plots). Table 3 presents the time-saving statistics for the multiclass case for MNIST-MNIST setup for the 5-layer custom model. We observe that the metric provides significant computational savings even for multiclass cases, with around $\times 50$ less time taken for *tl-frac*=0.01. However, the factor of improvement changes significantly with an increase in *tl-frac*, where for 4-class source to 3-class target, the benefit drops for $\times 51$ improvement to $\times 5$ (still good) when *tl-frac* goes from 0.01 to 0.05. Note that in Algorithm 1, at each step of the ternary search, the computation cost to calculate $\hat{P}^{tr}(q)$ is proportional to $q^{(m-1)}$ for $m$-ary source. Hence, for binary source, the cost is proportional to $q$ ($m$=2) but for 3-class and 4-class source, the cost scales as $q^2$ and $q^3$ respectively. Recall that the search space for $q^*$ depends on the $n^{val}$, which depends on dataset size (*tl-frac*). Hence, as *tl-frac* increases, the time improvement decreases non-linearly. More statistics on rank deviation for binary and multiclass cases are presented in Appendix D.1 and D.2.

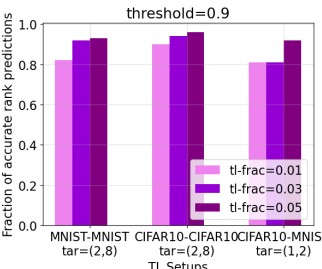 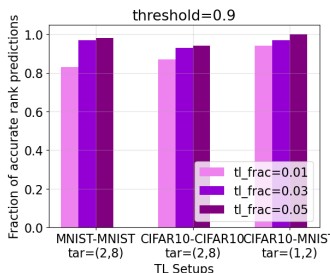 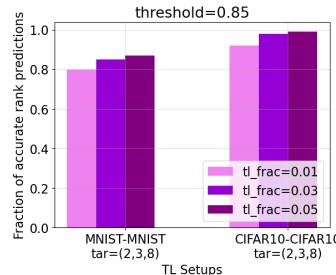

(a) 3-class source to binary target     (b) 4-class source to binary target     (c) 4-class source to ternary target

Figure 8: Rank similarity performance of metric for source tasks with high TL accuracy for varying data sizes (*tl-frac*) across TL setups using 5-layer custom model in multiclass case.

Table 3: Improvement in time taken for metric vs target neural network training for MNIST-MNIST TL Setup for 5-layer custom model for multi-class case. Values given in CPU seconds.

|  | **3-class Src, 2-class Tar Tar=(2,8)** | | | **4-class Src, 2-class Tar Tar=(2,8)** | | | **4-class Src, 3-class Tar Tar=(2,3,8)** | | |
| tl-frac | NN | Metric | Eff. | NN | Metric | Eff. | NN | Metric | Eff. |
|---|---|---|---|---|---|---|---|---|---|
| 0.01 | 5.24 | 0.11 | $\times\mathbf{48}$ | 4.75 | 0.10 | $\times\mathbf{47}$ | 6.41 | 0.13 | $\times\mathbf{51}$ |
| 0.03 | 11.19 | 0.21 | $\times\mathbf{53}$ | 8.04 | 0.32 | $\times\mathbf{25}$ | 11.59 | 0.71 | $\times\mathbf{16}$ |
| 0.05 | 12.03 | 0.32 | $\times\mathbf{38}$ | 11.53 | 0.75 | $\times\mathbf{15}$ | 15.85 | 2.89 | $\times\mathbf{5}$ |

## 6 CONCLUSION AND FUTURE WORK

In this paper, we studied the problem of selecting best pre-trained source model for transfer learning for a given target task with limited data. We propose BeST, a novel quantization-based task-similarity metric to measure transferability without needing a classical training process. The experimental results on different datasets show that the metric can accurately rank and predict the best pre-trained source model from a given group of models. It is shown that the metric is indifferent to the architecture of the custom model used until all of them have near-optimal performance. The performance of our metric increases with an increase in the number of data samples. We also show that our metric provides significant time savings over training a neural network for transfer learning implementation. There are certain limitations when scaling to multiclass cases where the computation time increases non-linearly with an increase in the number of data samples for transfer from source classifying a large number of classes. A possible future direction is to refine the method to overcome scalability challenges.

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

## A  PROOF OF THEOREM 4.1

*Proof.* For binary classifiers, the softmax output of the source model can be represented as $\mathbf{p} = [p_1, p_2]$ and since $p_1 + p_2 = 1$, we can only use $p_2$ to quantize $\mathbf{p}$ and represent as a one-hot vector. Here, quantization to level $q$ is intuitively equal to dividing the interval [0,1] into $q$ parts and indicating the bin where $p_2$ falls through a one-hot vector. Let $f_1 = f_{(p_2|Y=1)}$ and $f_2 = f_{(p_2|Y=2)}$ denote true underlying continous time conditional distribution of random variables $(p_2|Y = 1)$ and $(p_2|Y = 2)$ respectively. Denote $P_{i,j}(q) = P(\bar{X}_q = i|Y = j)$, $i \in \{1, .., q\}$ and $j \in \{1, 2\}$, where $P(q)$ is the true conditional probability distribution of random variable $(\bar{X}_q|Y)$ . Then $P_{i,1}(q)$ and $P_{i,2}(q)$ is given by Equation 6.

$$P_{i,1}(q) = \int_{(i-1)/q}^{i/q} f_1(x)\,dx, \ P_{i,2}(q) = \int_{(i-1)/q}^{i/q} f_2(x)\,dx \tag{6}$$

Assume that $\hat{P}^{tr}(q)$ and $\hat{P}^{val}(q)$ represent the estimation of $P(q)$ using the dataset $\mathcal{D}_q^{tr}$ and $\mathcal{D}_q^{val}$ calculated as – $\hat{P}_{i,j}^{tr}(q) = N_{i,j}^{tr}(q)/n^{tr}$; $\hat{P}_{i,j}^{val}(q) = N_{i,j}^{val}(q)/n^{val}$, where $N_{i,j}^{tr}(q)$ and $N_{i,j}^{val}(q)$ represent number of samples with label $Y = j$ for which the one-hot quantized vector has 1 at index $i$ using $\mathcal{D}^{tr}$ and $\mathcal{D}^{val}$ respectively. Using the definition of $N^{tr}(q)$ and $N^{val}(q)$, we can say $N_{i,j(q)}^{tr} \sim \text{Bin}(n^{tr}, P_{i,j}(q))$ and $N_{i,j}^{val}(q) \sim \text{Bin}(n^{val}, P_{i,j}(q))$, we have $E[\hat{P}_{i,j}^{tr}(q)] = P_{i,j}(q)$ and $E[\hat{P}_{i,j}^{val}(q)] = P_{i,j}(q)$.

For binary case, we can modify Equation 3 and 4 to Equation 7 and 8. Here $\pi_{*,i}^q = r$ means that for row $i$, $\hat{P}_{i,1}^{tr}(q) = \hat{P}_{i,2}^{tr}(q)$ or equivalently $N_{i,1}^{tr}(q) = N_{i,2}^{tr}(q)$.

$$A^{tr}(\boldsymbol{\pi}_*^q) = \max_{\boldsymbol{\pi}^q \in \mathcal{F}_q} \frac{1}{2} \sum_{i=1}^{q} \hat{\Pr}(\bar{X}_q^{tr} = i|Y^{tr} = \pi_i^q) = \frac{1}{2} \sum_{i=1}^{q} \max_{\pi_i^q} \hat{P}_{i,\pi_i^q}^{tr} \tag{7}$$

$$\pi_{*,i}^q = \begin{cases} \underset{j}{\text{argmax}} \ \{\hat{P}_{i,1}^{tr}(q), \hat{P}_{i,2}^{tr}(q)\} & \text{; if unique argmax exists} \\ r & \text{; if there both columns are equal} \end{cases} \tag{8}$$

It is given that the densities $f_1(x)$ and $f_2(x)$ are bounded i.e. $f_1(x), f_2(x) \leq B$. This means $0 \leq P_{i,1}, P_{i,2} \leq B/q$. For simplicity of notation, we write $A^{val}(\boldsymbol{\pi}_*^q)$ as $A^{val}(q)$, showing dependency on $q$. Using the definition to calculate $A^{val}(q)$, $E[A^{val}(q)]$ for dataset $\mathcal{D}^{val}$ can be expressed as a sum of two terms $A_1^{val}(q)$ and $A_2^{val}(q)$ as in Equation 9, where $A_1^{val}(q)$ and $A_2^{val}(q)$ are contributions from rows with $\hat{P}_{i,1}^{tr}(q) \neq \hat{P}_{i,2}^{tr}(q)$ and rows with $\hat{P}_{i,1}^{tr}(q) = \hat{P}_{i,2}^{tr}(q)$ respectively. We know that $0 \leq A^{val}(q), A_1^{val}(q), A_2^{val}(q) \leq 1$.

$$E[A^{val}(q)] = A_1^{val}(q) + A_2^{val}(q) \tag{9}$$

$A_1^{val}(q)$ and $A_2^{val}(q)$ can be expressed as in Equation 10 and 11.

$$A_1^{val}(q) = \frac{1}{2} \left\{ \sum_{i=1}^{q} \hat{P}_{i,\pi_{*,i}^q}^{val} \cdot \mathbb{1}_{(\pi_{*,i}^q \neq r)} \right\} \tag{10}$$

$$A_2^{val}(q) = \frac{1}{4} \left\{ \sum_{i=1}^{q} (\hat{P}_{i,1}^{val} + \hat{P}_{i,2}^{val}) \cdot \mathbb{1}_{(\pi_{*,i}^q = r)} \right\} \tag{11}$$

Since $A_1^{val}(q)$ is non-negative, we can use the markov inequality to write for any $\epsilon > 0$ -

$$\Pr(A_1^{val}(q) > \epsilon/2) \leq \frac{E[A_1^{val}(q)]}{\epsilon/2} \tag{12}$$

Focusing on the $E[A_1^{val}(q)]$ -

$$E[A_1^{val}(q)] = \frac{1}{2} E\left(\sum_{i=1}^{q} \hat{P}_{i,\pi_{*i}^q}^{val} \cdot \mathbb{1}_{(\pi_{*,i}^q \neq r)}\right)$$

$$= \frac{1}{2} \sum_{i=0}^{q-1} E\left(\hat{P}_{i,\pi_{*,i}^q}^{val} \cdot \mathbb{1}_{(\pi_{*,i}^q \neq r)}\right)$$

$$= \frac{1}{2} \sum_{i=1}^{q} \left(P_{i,1} \Pr(\pi_{*,i}^q = 1) + P_{i,2} \Pr(\pi_{*,i}^q = 2)\right)$$

$$= \frac{1}{2} \sum_{i=1}^{q} \left(P_{i,1} \Pr(N_{i,1}^{tr} > N_{i,2}^{tr}) + P_{i,2} \Pr(N_{i,1}^{tr} < N_{i,2}^{tr})\right) \tag{13}$$

For sake of simplicity we will denote $n^{tr}$ i.e. the number of samples for target training data, as simply $n$ as we do not need $n^{val}$ in the proof. We need to find upper bounds on $\Pr(N_{i,0}^{tr} > N_{i,1}^{tr})$ and $\Pr(N_{i,1}^{tr} < N_{i,2}^{tr})$ as given below.

$$\Pr(N_{i,1}^{tr} > N_{i,2}^{tr}) = \sum_{k=0}^{n-1} \Pr(N_{i,1}^{tr} > k) \Pr(N_{i,2}^{tr} = k))$$

$$\leq \Pr(N_{i,1}^{tr} > 0) \sum_{k=0}^{n-1} \Pr(N_{i,2}^{tr} = k))$$

$$= (1 - \Pr(N_{i,1}^{tr} = 0))(1 - \Pr(N_{i,2}^{tr} = n))$$

$$\leq (1 - \Pr(N_{i,1}^{tr} = 0))$$

$$= (1 - (1 - P_{i,1})^n)$$

$$\leq \left(1 - \left(1 - \frac{B}{q}\right)^n\right) \tag{14}$$

Similarly for $\Pr(N_{i,1}^{tr} < N_{i,2}^{tr})$ -

$$\Pr(N_{i,2}^{tr} > N_{i,1}^{tr}) = \sum_{k=0}^{n-1} \Pr(N_{i,2}^{tr} > k) \Pr(N_{i,1}^{tr} = k))$$

$$\leq \Pr(N_{i,2}^{tr} > 0) \sum_{k=0}^{n-1} \Pr(N_{i,1}^{tr} = k))$$

$$= (1 - \Pr(N_{i,2}^{tr} = 0))(1 - \Pr(N_{i,1}^{tr} = n))$$

$$\leq (1 - \Pr(N_{i,2}^{tr} = 0))$$

$$= (1 - (1 - P_{i,2})^n)$$

$$\leq \left(1 - \left(1 - \frac{B}{q}\right)^n\right) \tag{15}$$

Using upper bound on $P_{i,1}, P_{i,2}$ and equations 14 and 15 we can write -

$$E[A_1^{val}(q)] \leq \frac{1}{2} \sum_{i=1}^{q} \left(\frac{B}{q}\left(1 - \left(1 - \frac{B}{q}\right)^n\right) + \frac{B}{q}\left(1 - \left(1 - \frac{B}{q}\right)^n\right)\right)$$

$$= B\left(1 - \left(1 - \frac{B}{q}\right)^n\right) \tag{16}$$

Using equation 16 in equation 12 we get -

$$\Pr(A_1^{val}(q) > \epsilon/2) \leq \frac{2B\left(1 - \left(1 - \frac{B}{q}\right)^n\right)}{\epsilon} \tag{17}$$

Now for any given $\delta > 0, \forall q > Q(\frac{\epsilon}{2}, \frac{\delta}{2})$ where $Q(\epsilon, \delta) = \dfrac{B}{\left(1 - \left(1 - \frac{\epsilon\delta}{B}\right)^{1/n}\right)}$, we have -

$$\Pr(A_1^{val}(q) > \epsilon/2) \leq \frac{\delta}{2} \tag{18}$$

Now to get similar inequality on $A_2^{val}(q)$, define $A_3^{val}(q) = 1/2 - A_2^{val}(q)$. Since $A_3^{val}(q)$ is non-negative we can use markov inequality to write for any $\epsilon > 0$.

$$\Pr(A_3^{val}(q) > \epsilon/2) \leq \frac{E[A_3^{val}(q)]}{\epsilon/2} \tag{19}$$

Focusing on the $E[A_3^{val}(q)]$ -

$$
\begin{aligned}
E[A_3^{val}(q)] &= \frac{1}{4}E\left(2 - \sum_{i=1}^{q}(\hat{P}_{i,1}^{val} + \hat{P}_{i,2}^{val}) \cdot \mathbb{1}_{(\pi_{*,i}^q = r)}\right) \\
&= \frac{1}{4}E\left(\sum_{i=1}^{q}(\hat{P}_{i,1}^{val} + \hat{P}_{i,2}^{val})(1 - \mathbb{1}_{(\pi_{*,i}^q = r)})\right) \\
&= \frac{1}{4}\sum_{i=1}^{q}E\left((\hat{P}_{i,1}^{val} + \hat{P}_{i,2}^{val})(1 - \mathbb{1}_{(\pi_{*,i}^q = r)})\right) \\
&= \frac{1}{4}\sum_{i=1}^{q}(P_{i,1} + P_{i,2})\Pr(\pi_{*,i}^q \neq r) \\
&= \frac{1}{4}\sum_{i=1}^{q}(P_{i,1} + P_{i,2})(\Pr(N_{i,1}^{tr} > N_{i,2}^{tr}) + \Pr(N_{i,1}^{tr} < N_{i,2}^{tr})) \\
&\leq \frac{1}{4}\sum_{i=1}^{q}\left(\frac{2B}{q}\left(\left(1 - \left(1 - \frac{B}{q}\right)^n\right) + \left(1 - \left(1 - \frac{B}{q}\right)^n\right)\right)\right) \\
&= B\left(1 - \left(1 - \frac{B}{q}\right)^n\right) \tag{20}
\end{aligned}
$$

Now for any given $\delta > 0, \forall q > Q(\frac{\epsilon}{2}, \frac{\delta}{2})$ where $Q(\epsilon, \delta) = \dfrac{B}{\left(1 - \left(1 - \frac{\epsilon\delta}{B}\right)^{1/n}\right)}$, we have -

$$\Pr(A_3^{val}(q) > \epsilon/2) \leq \frac{\delta}{2} \tag{21}$$

Now combining equations 18 and 21 we can write, for any $\epsilon, \delta > 0, \forall q > Q(\frac{\epsilon}{2}, \frac{\delta}{2}) = \dfrac{B}{\left(1 - \left(1 - \frac{\epsilon\delta}{4B}\right)^{1/n}\right)}$, we can write -

$$\Pr(|E[A^{val}(q)] - \frac{1}{2}| > \epsilon) = \Pr(|A_1^{val}(q) + A_2^{val}(q) - \frac{1}{2}| > \epsilon)$$

$$\leq \Pr(|A_1^{val}(q) - 0| + |A_2^{val}(q) - \frac{1}{2}| > \epsilon)$$

$$\leq \Pr(|A_1^{val}(q) - 0| > \epsilon/2) + + \Pr(|A_2^{val}(q) - \frac{1}{2}| > \epsilon/2)$$

$$= \Pr(A_1^{val}(q) > \epsilon/2) + \Pr(\frac{1}{2} - A_2^{val}(q) > \epsilon/2)$$

$$= \Pr(A_1^{val}(q) > \epsilon/2) + \Pr(A_3^{val}(q) > \epsilon/2)$$

$$= \frac{\delta}{2} + \frac{\delta}{2}$$

$$= \delta \tag{22}$$

**Remark.** *Since we are dealing with probabilities, for $\delta > 1$ equation 22 will always be satisfied for any q. Hence, practically we deal with $0 < \delta \leq 1$. Also, in the proof we used the fact that $\epsilon\delta \leq 4B$ so that $Q(\frac{\epsilon}{2}, \frac{\delta}{2})$ is well defined and we don't take $n^{th}$ root of a negative number.*

$\square$

## B  ARCHITECTURES FOR SOURCE AND CUSTOM MODEL

The neural network architectures used to build the CNN for the source models trained on MNIST and CIFAR10 are given in Table B.1 and B.2 respectively. The 2-layer and 5-layer architecture used for the custom model is given in Table B.3.

Table B.1: CNN architecture for Source Tasks using MNIST.

| Layer Type | Parameters | Activation |
|---|---|---|
| Conv2D 1 | filters=32, kernel=(3x3) | relu |
| Max-Pooling 1 | pool-size=(2x2) | - |
| Conv2d 2 | filters=64, kernel=(3x3) | relu |
| Max-Pooling 2 | pool-size=(2x2) | - |
| Flatten | - | - |
| Dropout | probability=0.5 | - |
| Output | classes=m | softmax |

Table B.2: CNN architecture for Source Tasks using CIFAR-10.

| Layer Type | Parameters | Activation |
|---|---|---|
| Conv2D 1 | filters=32, kernel=(3x3) | relu |
| Conv2D 2 | filters=32, kernel=(3x3) | relu |
| Max-Pooling 1 | pool-size=(2x2) | - |
| Conv2d 3 | filters=64, kernel=(3x3) | relu |
| Conv2d 4 | filters=64, kernel=(3x3) | relu |
| Max-Pooling 2 | pool-size=(2x2) | - |
| Flatten | - | - |
| Dense | neurons=128 | relu |
| Output | classes=m | softmax |

## C  EXPERIMENTAL SETTINGS

As explained in Section 5, for each target task for any of the 3 TL setups, we consider 45 different pre-trained source models. The specifics of which 45 source models are selected for each TL setup

Table B.3: DNN architecture for Transfer Learning Models.

| Layer Type | Neurons | Activation |
|---|---|---|
| **2-layer** | | |
| Dense | 10 | relu |
| Output | n | softmax |
| **5-layer** | | |
| Dense | 10 | relu |
| Dense | 20 | relu |
| Dense | 40 | relu |
| Dense | 10 | relu |
| Output | n | softmax |

for source classifiers with different numbers of classes are explained below. Recall that both MNIST and CIFAR10 have 10 classes.

**Binary source models:** When the source model is a binary classifier, we choose all possible unique combinations of choosing 2 classes out of 10 to define the source models. Note that classifying data of classes 1 and 5 and 5 and 1 are considered the same source model. Let Src = $(i, j)$ denote that the source model is trained to classify images from classes indexed $i$ and $j$ where $i, j \in \{1, 2, .., 10\}$ and $i \neq j$ (e.g., Src=(1,2) for MNIST means source model to classify digits 0 and 1 as the $1^{st}$ class corresponds to digit 0 and so on). Then the set of 45 source models can be represented as a set of tuples $S = \{(1, 2), (1, 3), ....(2, 3), (2, 4), ..., (9, 10)\}$.

**3-class source models:** Let Src = $(i, j, k)$ denote that the source model is trained to classify images from classes indexed $i, j$ and $k$ where $i, j, k \in \{1, 2, .., 10\}$ and $i \neq j \neq k$ (e.g., Src=(1,2,3) for MNIST means source model to classify digits 0, 1 and 2). When the source model is a ternary classifier, there are more than 45 unique combinations of choosing 3 classes out of 10. We choose the first 45 unique combinations of 3 classes defined in the order given by set $S = \{(1, 2, 3), (1, 2, 4), ..., (2, 3, 4), (2, 3, 5), ...(2, 4, 6)\}$.

**4-class source models:** Let Src = $(i, j, k, l)$ denote that the source model is trained to classify images from classes indexed $i, j, k$ and $l$ where $i, j, k, l \in \{1, 2, .., 10\}$ and $i \neq j \neq k \neq l$ (e.g., Src=(1,2,3,4) for MNIST means source model to classify digits 0, 1, 2 and 3). We choose the first 45 unique combinations of 4 classes out of 10, defined in the order given by set $S = \{(1, 2, 3, 4), (1, 2, 3, 5), ..., (1, 3, 4, 5), (1, 3, 4, 6), ...(1, 3, 7, 9)\}$.

# D    RANK DEVIATION STATISTICS

## D.1    BINARY CASE

The statistics on the deviation of the predicted rank from the true ranks similar to Table 1 is presented for CIFAR10-CIFAR10 and CIFAR10-MNIST TL setups in Table D.1 and D.2 respectively. We can observe that for a threshold of 0.8 i.e. ranking source models with a transfer learning accuracy of $> 80\%$, the average deviation of ranks is less than 1 rank for CIFAR10-CIFAR10 setup and less than 2 ranks for CIFAR10-MNIST respectively. This holds across different data sizes, varying from as low as $\sim 100$ samples (*tl-frac*=0.01) to $\sim 500$ samples (*tl-frac*=0.05), in both 2-layer and 5-layer custom model architectures.

Table D.1: Statistics for deviation of predicted ranks from true ranks for CIFAR10-CIFAR10 TL Setup with target task to classify images of aeroplane and automobile.

| TL Model | Threshold | tl-frac=0.01 | | | tl-frac=0.03 | | | tl-frac=0.05 | | |
|---|---|---|---|---|---|---|---|---|---|---|
| | | 0.5 | 0.7 | 0.8 | 0.5 | 0.7 | 0.8 | 0.5 | 0.7 | 0.8 |
| 2-layer | Mean | 6.11 | 1.56 | **0.31** | 5.45 | 1.64 | **0.14** | 6.55 | 1.65 | **0.14** |
| | Std | 1.28 | 0.72 | 0.40 | 1.58 | 0.72 | 0.24 | 1.43 | 0.93 | 0.24 |
| 5-layer | Mean | 7.47 | 1.81 | **0.34** | 6.19 | 1.55 | **0.15** | 5.93 | 1.43 | **0.13** |
| | Std | 2.10 | 0.65 | 0.30 | 2.13 | 0.8 | 0.27 | 1.55 | 0.63 | 0.23 |

Table D.2: Statistics for deviation of predicted ranks from true ranks for CIFAR10-MNIST TL Setup with target task to classify images of digits 0 and 1.

| TL Model | Threshold | tl-frac=0.01 | | | tl-frac=0.03 | | | tl-frac=0.05 | | |
|---|---|---|---|---|---|---|---|---|---|---|
| | | 0.5 | 0.7 | 0.8 | 0.5 | 0.7 | 0.8 | 0.5 | 0.7 | 0.8 |
| 2-layer | Mean | 4.90 | 1.45 | **0.50** | 3.76 | 1.46 | **0.56** | 2.92 | 1.05 | **0.41** |
| | Std | 1.06 | 0.69 | 0.50 | 0.9 | 0.59 | 0.47 | 1.42 | 0.66 | 0.57 |
| 5-layer | Mean | 5.24 | 1.93 | **0.83** | 3.62 | 1.56 | **0.44** | 3.38 | 1.29 | **0.26** |
| | Std | 1.12 | 0.49 | 0.46 | 1.28 | 0.67 | 0.50 | 1.55 | 0.95 | 0.41 |

## D.2 MUTLICLASS CASE

Similar to the binary case, we present the rank deviation statistics (mean and standard deviation) for MNIST-MNIST TL setup for 3-class source to 2-class target, 4-class source to 2-class target, and 4-class source to 3-class target in Table D.3. As observed in the binary case, the metric performs best for source models with high transfer learning accuracy i.e. high *threshold* (highlighted in bold). For *threshold*=0.9, the mean rank deviation is less than 1 rank which shows the ability of our metric to rank the source well across different multiclass transfer learning settings. The mean rank deviation also decreases with an increase in dataset size (*tl-frac*) as expected. We can also observe that the standard deviation of the rank deviations is less than 2 i.e. individual rank deviations are not too far from the average.

Table D.3: Statistics for deviation of predicted ranks from true ranks for MNIST-MNIST TL Setup for multiclass case.

| tl-frac | Threshold | 3-class to 2-class Tar=(2,8) | | | 4-class to 2-class Tar=(2,8) | | | 4-class to 3-class Tar=(2,3,8) | | |
|---|---|---|---|---|---|---|---|---|---|---|
| | | 0.5 | 0.7 | 0.9 | 0.5 | 0.7 | 0.9 | 0.5 | 0.7 | 0.9 |
| 0.01 | Mean | 7.49 | 5.99 | **1.01** | 5.89 | 5.22 | **1.57** | 7.42 | 4.25 | **0.15** |
| | Std | 1.99 | 1.73 | 0.92 | 1.70 | 1.53 | 1.39 | 2.03 | 1.05 | 0.26 |
| 0.03 | Mean | 4.24 | 4.18 | **0.60** | 3.82 | 3.81 | **0.45** | 4.99 | 3.69 | **0.03** |
| | Std | 1.23 | 0.95 | 0.51 | 0.85 | 0.83 | 0.47 | 1.51 | 1.29 | 0.12 |
| 0.05 | Mean | 2.90 | 2.84 | **0.47** | 3.09 | 3.09 | **0.29** | 4.8 | 3.42 | **0.07** |
| | Std | 0.98 | 0.94 | 0.34 | 0.79 | 0.79 | 0.43 | 1.83 | 1.72 | 0.21 |

## E STATISTICS TO JUSTIFY UNIMODAL APPROXIMATION

Table E.1: Statistics to compare the difference for MNIST-MNIST TL setup using 5-layer custom model.

| tl-frac | Stats | 2-class Src to 2-class Tar Tar=(2,4) | 3-class Src to 2-class Tar Tar=(2,8) |
|---|---|---|---|
| 0.01 | Mean | **0.041** | **0.045** |
| | Std | 0.021 | 0.025 |
| 0.03 | Mean | **0.024** | **0.027** |
| | Std | 0.014 | 0.014 |
| 0.05 | Mean | **0.017** | **0.019** |
| | Std | 0.011 | 0.010 |

One of the key aspects of our algorithm design is the unimodal approximation for variation of $A^{val}(\pi_*^q)$ vs $q$ in the search space $q \in (2, n^{val}/n)$. This approximation allows us to avoid calculating $A^{val}(\pi_*^q)$ at every $q$ and instead use ternary search to find the maximum $A^{val}(\pi_*^q)$. To justify the approximation, we present the mean and standard deviation of the absolute value of the difference in calculated metric (i.e. the maximum $A^{val}(\pi_*^q)$) using brute force search and ternary search in Table E.1. Observe that the mean difference (in bold) for 2-class source to 2-class target and 3-class source to 2-class target has a maximum value of $\sim 0.05$ for *tl-frac*=0.01 (100 samples

case). This means on average, the best $A^{val}(\pi_*^q)$ found by ternary search differs at max by 5% as compared to the one calculated by brute-force. Observe that the mean decreases with an increase in dataset size (*tl-frac*) with a value as low as $< 0.02$ (i.e. $< 2\%$ accuracy difference) for 500 samples (*tl-frac*=0.05). The small standard deviation suggests that the actual difference is close to the mean difference. The low average difference suggests that the unimodal approximation is a reasonable approximation as ternary search gives a value very close to the brute-force value while giving significant time savings.

# F    CORRELATION STATISTICS

In addition to the evaluation of our metric in Section 5 using the fraction of correct ranks as illustrated in Figures 6, 7 and 8, in this section we also evaluate the performance of our metric using standard statistical correlation coefficients namely - Pearson, Spearman and Kendall's Tau.

Figure F.1, F.2 and F.3 presents the correlation values for Pearson, Spearman, and Kendall's Tau correlation coefficients respectively, as dataset size (*tl-frac* parameter) changes. Each figure has 4 subfigures, representing the correlation data for 4 cases - 1) 2-class Source to 2-class Target, 2) 3-class Source to 2-class Target, 3) 4-class Source to 2-class Target, and 4) 4-class Source to 3-class Target. Within each subfigure, the legend, 'CIFAR10-MNIST Tar=(1,2)' for example, indicates the transfer learning setup where the source model has been trained to classify images from CIFAR10 while the target task is to learn to classify images from MNIST belonging to the $1^{st}$ and $2^{nd}$ classes (i.e. images of digits 0 and 1).

As explained in Section 5, for each target task, we use 45 unique pre-trained source models and for each source, we evaluate the transfer learning performance after training and calculate our metric using Algorithm 1. The correlation statistics presented are the average of the correlation values over 20 runs, where for each run, the correlation is measured between two lists of 45 items each (as there are 45 sources).

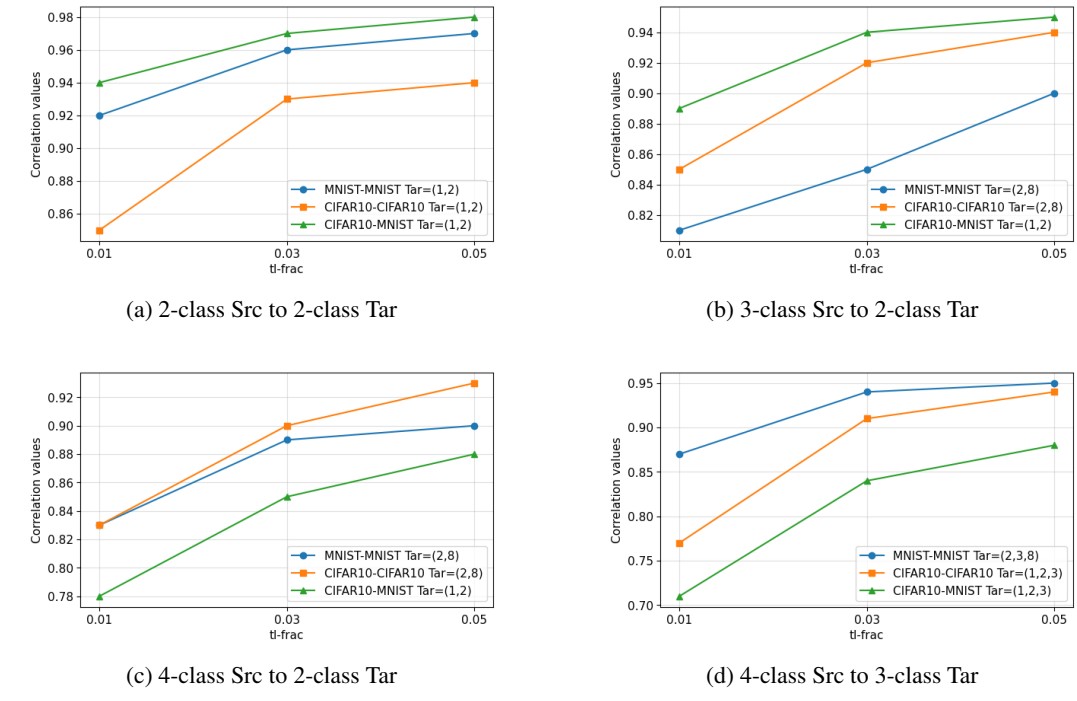

(a) 2-class Src to 2-class Tar

(b) 3-class Src to 2-class Tar

(c) 4-class Src to 2-class Tar

(d) 4-class Src to 3-class Tar

Figure F.1: Pearson Correlation values

The Pearson coefficient measures the correlation directly between the transfer accuracy and the metric values. However, Spearman and Kendall's Tau coefficient measures the correlation between the rankings deduced from these values and not the values directly. Recall that our metric aimed to rank the source models accurately according to their transferability for a particular target task.

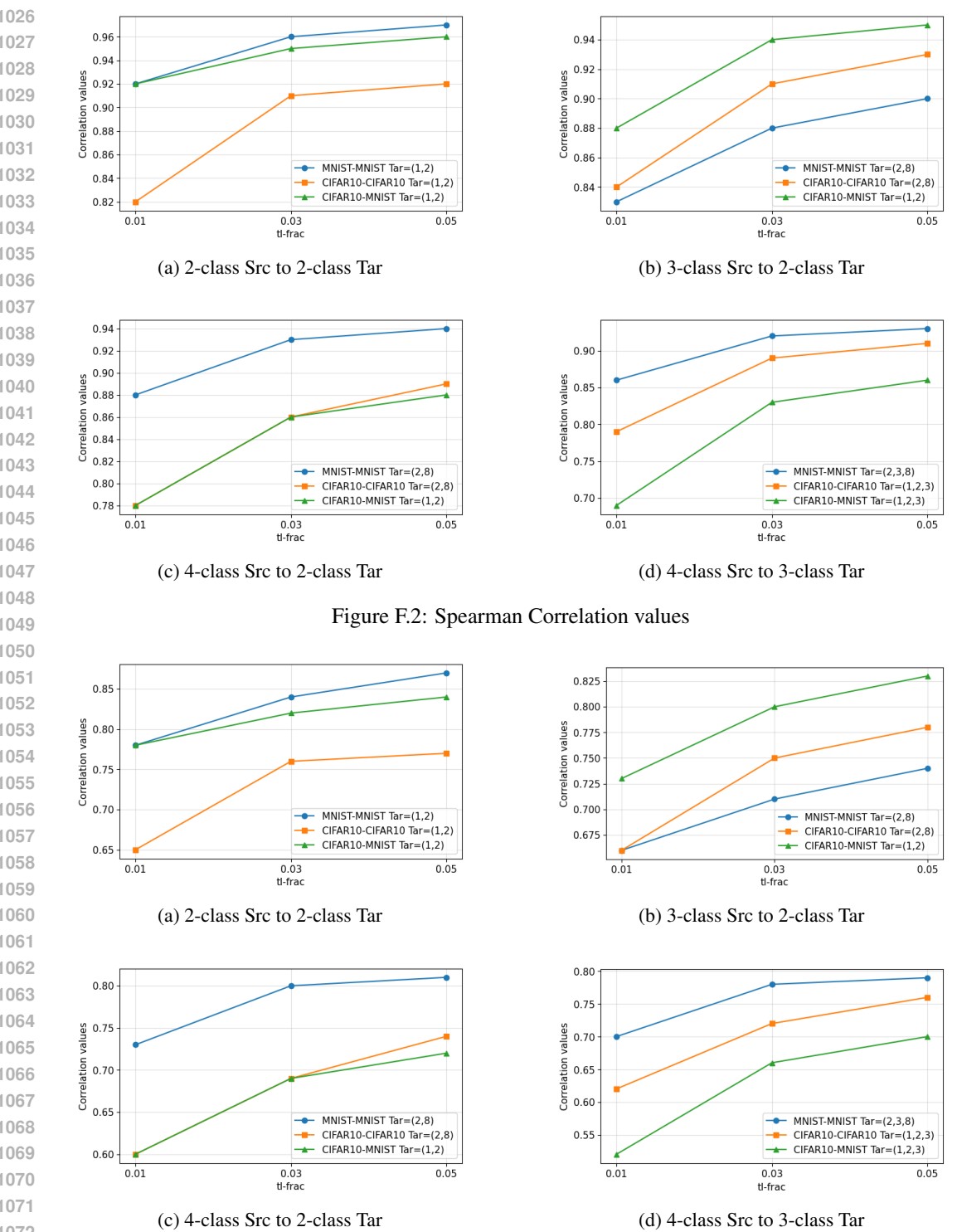

Figure F.2: Spearman Correlation values

Figure F.3: Kendall's Tau Correlation values

From all these figures, we can observe that the correlation values for all three correlation methods are high indicating that our metric values and the rankings provided are highly correlated with the transfer learning accuracy. We can also observe that the correlation values increase as the dataset size increases (*tl-frac* parameter goes from 0.01 to 0.05). This is in line with our expectation as performance improves with the availability of more data.

## G  EXPERIMENTS WITH IMAGENETTE DATASET

Imagenette Howard (2019) is a smaller, curated subset of the ImageNet dataset designed to speed experiments and make them more accessible for research and practice. It contains only 10 classes of images: tench, English springer, cassette player, chain saw, church, French horn, garbage truck, gas pump, golf ball, and parachute. The images used are $64 \times 64$ pixels. We can call this setup the 'Imagenette-Imagenette' transfer learning setup (nomenclature similar to MNIST-MNIST, CIFAR10-CIFAR10, etc.), where both the source and target models are to classify images from the Imagenette dataset.

We present the results for a binary classification case in which both source and target models are binary classifiers in the Imagenette dataset, trying to classify images from 2 out of the 10 classes in the dataset. The source model architecture used is the same as that used for CIFAR10 (Table B.2) and the 5-layer custom model (as in Table B.3) is used to build and train the target model. To demonstrate the variation of results with dataset size, we used 3 *tl-frac* values - 0.1, 0.3, and 0.5 (corresponding to $\sim 150$, $\sim 450$, and $\sim 750$ samples). We used 10 runs for each source-target pair (each run has a different subset of target data) instead of the 20 runs used for the earlier setups.

### G.1  VISUALIZATION OF COMPARISON OF RANK BY METRIC VS TRUE RANK

Similar to Figure 5, the comparison of ranks given by the true transfer performance (test accuracy after training target model using a particular source), metric calculated using brute force, and metric calculated using our ternary search approach, as given in Algorithm 1, is presented in Figure G.1 using $\sim 150$ samples and Figure G.2 using $\sim 750$ samples[4]. Tar=(1,2) indicates that the target task here is to classify images from the $1^{st}$ and $2^{nd}$ classes in Imagenette (*tench* and *English springer*).

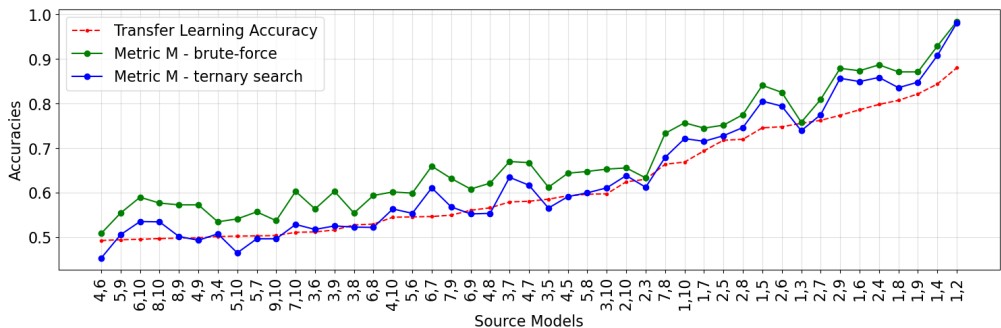

Figure G.1: Comparison of ranks predicted by metric and ground truth for 2-class source to 2-class target transfer in Imagenette-Imagenette transfer setup (Tar=(1,2)) using $\sim 150$ data samples.

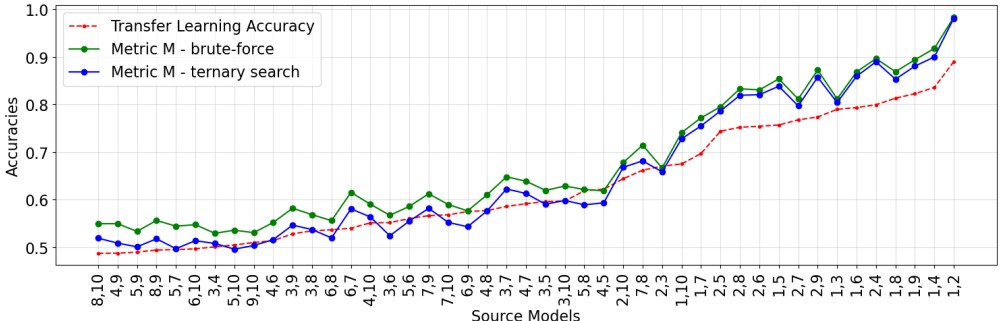

Figure G.2: Comparison of ranks predicted by metric and ground truth for 2-class source to 2-class target transfer in Imagenette-Imagenette transfer setup (Tar=(1,2)) using $\sim 750$ data samples.

---

[4]Observe that the order of source models (x-axis) change with the change is the size of the dataset as the transfer learning accuracy changes.

The x-axis contains the source models represented by a tuple where, for example, the '2,4' indicates that the source model is trained to classify images from $2^{nd}$ and $4^{th}$ class in Imagenette (*English springer* and *chain saw*). We omit the names of the classes to avoid crowding in the figures.

Again, we can observe a very small deviation between the metric value calculated using brute force (green) and the values calculated using ternary search (blue) in both figures. This results in the rankings given by the metric using ternary search being very similar to those given by the metric calculated using brute force (supporting our unimodal approximation)

If we look at the most transferable source for the target task used (Tar=(1,2)) in Figure G.1 and G.2 i.e. source to classify images from classes (1,2), (1,4), (1,9), etc., we can observe that all of them have one class in common with the target task. It makes intuitive sense that a source model that can classify images from classes indexed 1 and 4 should be able to classify images from classes 1 and 2 (target task) pretty well as it already knows how to classify images from class 1. The result also aligns with our intuition that the best transferable source should be the one that classifies the same classes as the target i.e. (1,2) here.

## G.2    FRACTION OF ACCURATE RANK PREDICTIONS

Similar to Figure 6, the fraction of correct ranks for different *threshold* values is presented in Figure G.3. We can observe that for both the target tasks (Tar=(1,2) in Figure G.3a and Tar(2,3) in Figure G.3b), as observed with the other datasets in Section 5, the fraction of correct ranks improve as we evaluate for higher *threshold* i.e. evaluate for source models with higher transfer learning accuracy. We can also observe that the fraction of correct ranks improves for a given *threshold* as the dataset size increases (*tl-frac* increases).

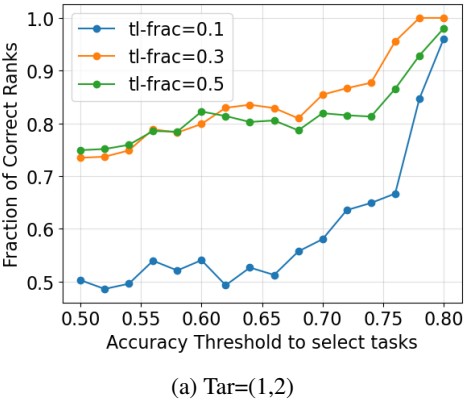 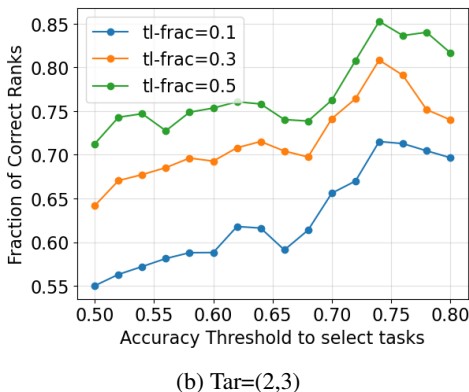

(a) Tar=(1,2)            (b) Tar=(2,3)

Figure G.3: Fraction of accurate ranks Vs *threshold* for Imagenette-Imagenette Transfer Setup for different dataset sizes using 5-layer custom model.

## G.3    TIME IMPROVEMENT STATISTICS

Table G.1: Improvement in time taken for metric calculation vs target neural network training for Imagenette-Imagenette TL setups for 5-layer custom model for binary classifier case. Values given in CPU seconds.

| tl-frac | Tar=(1,2) | | | Tar=(2,3) | | |
|---|---|---|---|---|---|---|
| | Training | Metric | Eff. | Training | Metric | Eff. |
| 0.1 | 46.35 | 0.94 | ×**48** | 48.69 | 0.96 | ×**51** |
| 0.3 | 112.98 | 2.64 | ×**42** | 115.73 | 2.66 | ×**43** |
| 0.5 | 180.84 | 4.35 | ×**41** | 178.19 | 4.37 | ×**41** |

The comparison of the time taken to train a target model using a pre-trained source to evaluate the transfer accuracy versus time taken to calculate the proposed transferability metric for the Imagenette-Imagenette transfer setup for two target tasks (Tar=(1,2) and Tar=(2,3)) is given in Table

G.1. Similar to the results in Table 2 and 3, we can observe that our calculating our metric provides significant time savings as compared to finding the transfer performance through training, with improvements (Eff. column) of around $\times\, 40$ and more. We can also observe that the time improvement is reflected for different dataset sizes (*tl-frac*). With an increase in dataset size, there is a decrease in time efficiency but the decrease is non-linear. The computational improvements offered by our metric for a larger dataset like Imagenette suggest that the results are scalable and not limited to smaller datasets like MNIST and CIFAR10.

# H  VARIATION WITH DIFFERENT RANDOM SEED FOR PRE-TRAINED SOURCE

The pre-trained source model is trained once for each source task to be used multiple times for different target data and target models. Hence, it is important to demonstrate that the results are not sensitive to a particular source network initialization parameter. In this section, we present results considering sources trained on 5 different random seeds.

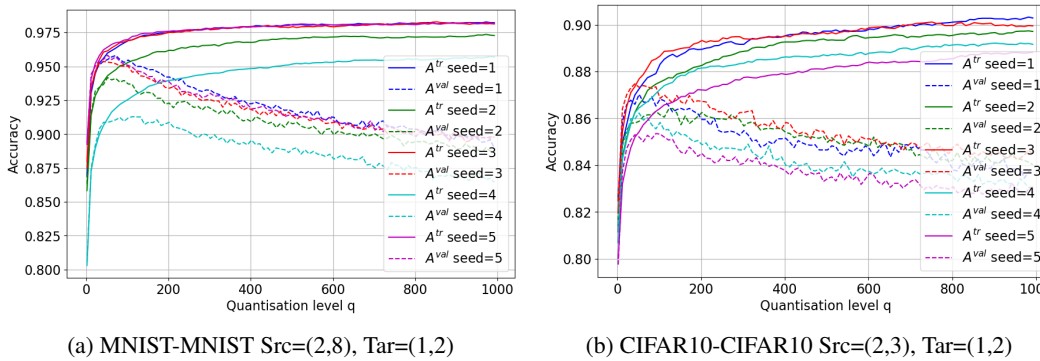

(a) MNIST-MNIST Src=(2,8), Tar=(1,2)        (b) CIFAR10-CIFAR10 Src=(2,3), Tar=(1,2)

Figure H.1: Train-validation accuracy tradeoff for 5 different random seeds used to initialize the source model before training. The source and target tasks are binary classifiers.

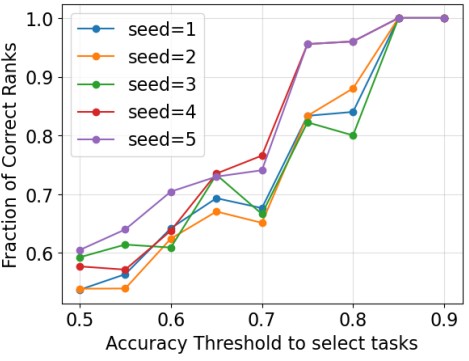
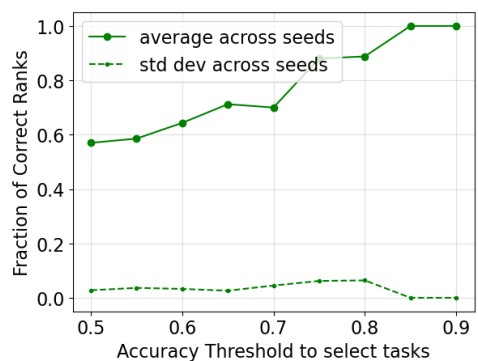

(a) Fraction correct ranks across different random seeds

(b) Average and Standard Deviation across different random seeds

Figure H.2: Train-validation accuracy tradeoff for 5 different random seeds used to initialize the source model before training. The source and target tasks are binary classifiers. The target task is to classify images from classes *airplane* and *automobile* (Tar=(1,2))

Similar to Figure 4, we present the variation of $A^{tr}(\pi_*^q)$ and $A^{val}(\pi_*^q)$ vs quantization level $q$ for 5 different random seeds in Figure H.1a and H.1b. Figure H.1a presents the variation for MNIST-MNIST transfer setup with source model to classify digits 1 and 7 (Tar=(2,8)[5]) while the target task is to classify digits 0 and 1 (Tar=(1,2)). Similarly, Figure H.1b presents the variation for CIFAR10-CIFAR10 transfer setup with source model to classify images from classes index 2 and 3 (Src=(2,3))

---

[5]Digit 1 is the $2^{nd}$ and 7 is the $8^{th}$ class in MNIST

and the target task is to classify images from classes index 1 and 2 (Tar=(1,2)) in CIFAR10. We can observe that the structure of the variations remains the same across different random seeds. We can also observe the same unimodal-like structure for all 5 random seeds.

Specifically, for the CIFAR10-CIFAR10 transfer setup using the 5-layer custom model, the fraction of accurate ranks vs *threshold* for different random seeds is presented in Figure H.2a. Figure H.2b presents the average and the standard deviation for each *threshold* value across the 5 random seeds. We can observe Figure H.2a that though there is some variation for values across random seeds (as expected), the overall trend is similar which implies that the results in Section 5 are not a consequence of a particular network initialization. From Figure H.2b, we can observe that the standard deviation of the fraction of correct ranks is not significant ($<5\%$) and the average fraction of correct rank predictions increase when we use a higher *threshold* i.e. select source models with higher transfer learning performance.

