# OpenReview forum: "BeST - A Novel Source Selection Metric for Transfer Learning"
_ICLR.cc/2025/Conference — Submitted to ICLR 2025_

### Official Review · Reviewer_g5gx · 2024-11-01

**Soundness:** 3
**Presentation:** 2
**Contribution:** 3
**Rating:** 6
**Confidence:** 4

**Summary:**

This submission presents a metric for selection of the best source tasks that may be adaptive to target task. The key lies in the selection of the most transferable sources for a given task. This work shows a quantization-level optimization method. Experiments show the metric performs well over different datasets.


The authors do not provide feedback. I therefore reduce my score.

---Since the authors have further provided feedback about my concerns and most of my concerns have been addressed. I keep my original rating, because the writting may have space to improve.

**Strengths:**

1. The topic for selecting most transferable source tasks for a given target task is interesting and sounds good.
2. The proposed method has some technical novelty and I like this idea during the numerous transfer learning methods.
3. It is good to fully consider the computational savings during selection of possible source models.

**Weaknesses:**

1. I think the very weakness lies in the writting of the technical parts. It is obscure and sometimes difficult to understand the mathematical part such as the policy optimization and computation. It is unclear why such computation is proposed, such as Eq.1 and lack intuition behind this idea.
2. I think in Algorithm 1, it may not be easy to reproduce from the current description.
3. I suggest the authors clarify some simple concept such as task-similarity, selection strategy, etc. in implementation, without particular wrap up in mathematical aspects.
4. This submission will have a higher quality if the writting is further improved. I lean to accept this submission, and expect the authors' feedback.

**Questions:**

I would like to ask what is the intuition behind the complex design of the metric?

---

> ### Author Response · Authors · 2024-11-27
> **Response for Weaknesses**
>
> We thank you for your valuable feedback and apologize for the delay in our response as we were trying to add new simulation results to respond to the comments by all the reviewers.
>
> > **I think the very weakness lies in the ... lack intuition behind this idea.**
>
> Response: We apologize if certain parts are too technical and will try to explain in simple words here.
>
> 1. Equation 1 deals with the quantization of a softmax vector to a $q$-level quantized vector. Figure 2 explains the process of converting a 3x1 softmax vector to a 9x1 quantized vector which involves mapping the 3x1 vector to a 2-D grid and then unfolding the grid in row-by-row fashion to get a column vector. Equation 1 is nothing but a fancy way to denote the index i’ in the final column vector that will be 1 while the others are 0.
> 2. In its simplest form, quantization of a vector [0.9, 0.1] with $q$=4 is simply dividing the number line [0,1] into 4 parts and deciding which part 0.1 belongs to (in this case, the vector will read [1,0,0,0]).
> 3. The entire idea as to why the quantization approach was introduced in the first place is explained in the next response (while answering the question in questions section of your review, adding as a separate response as word limit reached).
> 4. The key steps in the proposed method are - quantization, finding a policy for each quantization, finding train and validation accuracy in the analytical form, finding the optimal quantization, and calculating the corresponding metric. Having explained quantization earlier, we can focus on the policy. It is clear that there are multiple possible mappings $\pi^q$ from the quantized source output to the target labels. Each mapping has a train and validation accuracy. **Remember these train and validation accuracies here are not an output of traditional training but the performance of the analytical mapping**. We need to find a policy $\pi^q_*$ that maximizes the training accuracy (Eq 4). The variation of validation accuracy of this policy $\pi^q_*$ with quantization levels is given in Figure 4. Now, the objective is to find the quantization $q^*$ that gives the highest validation accuracy. The highest validation accuracy is our metric M (Eq 5).
>
> > **I think in Algorithm 1, it may not be easy to reproduce from the current description.**
>
> Response: We apologize if the description in Algorithm 1 is unclear.
> 1. Refer to Figure 4, where we can see the variation of $A^{val}$ with $q$. Algorithm 1 is simply a ternary search (variation of binary search) on the variable $A^{val}$ using a right and left pointer named $m_1$ and $m_2$.
> 2. In line 1, we first make sure that the dataset is balanced i.e. equal samples for each class, then the left and right search boundary is defined in line 2.
> 3. Lines 3-16 are just running basic ternary search to find the mode of $A^{val}$ vs $q$ and the steps inside the while loop is simply the steps to calculate $A^{val}$ for $q=m_1$ and $q=m_2$.
> 4. Once stopping condition is reached, lines 17-21 are simply steps to calculate $A^{val}$ at the final right and left pointers $L$ and $M$. The metric is the average of $A^{val}$ at these two pointers.
> 5. Note - Steps 5 and 17, where a quantized version of dataset is created, it simply means that we did a forward pass of the target data through the pre-trained source model and performed the quantization (with level $q$) of the softmax vector to get a one-hot vector.
>
> > **I suggest the authors clarify some simple concept such as task-similarity, selection strategy, etc. in implementation, without particular wrap up in mathematical aspects.**
>
> Response:
> 1. In simple words, task similarity depends on the capability of the source model to distinguish target data by providing different output representations for different labels. If the source output is similar for target data from different classes, the tasks are not very similar (when we talk about tasks, we mean models as task is an abstract concept). It also depends on the training parameters, transfer architecture, and target dataset. In our paper, we define task similarity or task transferability by the test accuracy achieved when the target model is trained using a frozen pre-trained source model using target data (Figure 1).
> 2. As explained while replying to weakness 1, for every quantization level $q$, there are multiple mapping policies $\pi^q$ possible that map the quantized source output to target labels. For each quantization level, we want to select a policy $\pi^q_*$ that maximizes the training accuracy and then calculate its validation performance. Finally, we want to search for the optimal quantization level $q^*$ that maximizes the validation accuracies across different such policies. The maximum validation accuracy calculated through this process is the proposed metric M (Eq 5). In implementation (Algo 1), we calculate the metric using a ternary search to find this optimal validation accuracy $A^{val}(\pi^{q^*}_*)$.

---

> > ### Author Response · Authors · 2024-11-27
> > **Response for Questions**
> >
> > > **I would like to ask what is the intuition behind the complex design of the metric?**
> >
> > Response: Thank you for the question. We would like to elaborate on the reasoning behind our current approach.
> > 1. The intuition behind the design of the metric can be explained in the form of a chain of thoughts. We started with trying to mathematically analyze the ability of the pretrained source model to differentiate the target data and considered a simple setup for a binary classification case where the source model would just output 0 or 1 (no softmax output, call this hardmax case). In this case, the added layers on top of the frozen source (Figure 1) perform a transformation of the label predicted by the source to the correct target label. It is easy to write an expression of the maximum accuracy any such transformation function can achieve using the conditional probability distribution of the source model output given the target label.
> > 2. Until here, we did not use the information hidden in the softmax output of the source model which, if nothing, should improve our performance. However, such a discrete mapping as in the hardmax case has the benefit that we can write the accuracy expressions easily which is not true for the softmax case. Hence, as a middle ground, we thought of this idea of quantization of softmax outputs to convert to multi-level discrete outputs where instead of having 2 states in the hardmax case (0 and 1), we can have $q$ states. We quickly understood that having a low quantization (like hardmax case i.e. $q=0$) and high quantization (like $q=10000$) are both suboptimal, as for low quantization, we are quantizing too harshly, while for high quantization, the small number of target samples creates imprecise distribution estimations. There might be an optimal quantization level for the given dataset that gives us the highest validation accuracy. This optimal $q^*$ and the corresponding metric M are represented in Equation 5.
> >
> > ---
> >
> > ## Additions to the paper in appendix section
> > Based on the comments of other reviewers, we have added new simulation results in the paper (described below) in the appendix section.
> > 1. **Appendix F - Correlation Statistics** - in addition to evaluating the source rankings provided by the metric vs true rankings using the fraction of correct rank predictions (in Section 5), we have added the correlation statistics using Spearman, Pearson and Kendall's Tau correlations.
> > 2. **Appendix G - Experiments with Imagenette Dataset** - we ran addition simulations for the Imagenette dataset (subset of Imagenet dataset) for 64x64 images and presented the results involving - visualization of the predicted vs true rankings, fraction of correct rank predictions, and time improvement statistics.
> > 3. **Appendix H - Variation with different random seed for pre-trained source** - as per the comment of one of the other reviewers, since the method depends on the pre-trained source model, we tried to run a few simulations with 5 different random seeds to initialize the pre-trained source model parameters before training. The results are presented in this section indicating that our results are not dependent on a particular network initialization.
> >
> > ---
> >
> > We would like to sincerely acknowledge the delay in our response, which was due to the time required to conduct additional simulations to address the reviewer’s feedback and provide new results. We noticed that the rating has decreased from 6 to 3, and we apologize if this reduction was influenced by our delayed submission.
> >
> > We hope that our detailed comments and the additional data have provided clarity on the questions and concerns raised regarding our paper. We humbly request the reviewer to kindly reconsider their evaluation in light of the updated information, should they find it appropriate.

---

### Official Review · Reviewer_mueD · 2024-11-02

**Soundness:** 2
**Presentation:** 1
**Contribution:** 2
**Rating:** 3
**Confidence:** 3

**Summary:**

The manuscript introduces a new metric for selecting source models in transfer learning, utilizing quantization to estimate transferability. It provides theoretical backing to guarantee performance improvements, particularly in binary classification tasks, offering a practical approach for enhancing model fine-tuning.

**Strengths:**

1. The paper addresses a highly relevant scenario in the current landscape of machine learning where fine-tuning a pre-trained model is more common and practical than training a model from scratch. This approach is valuable for efficiently leveraging existing computational resources and knowledge.
2. The methodology involving quantization to estimate transferability introduces a novel technique to the field of transfer learning.
3. The paper provides a theoretical foundation that ensures the performance of the proposed method, especially highlighted in simpler scenarios such as binary classification.

**Weaknesses:**

1. The experiments are conducted using only DNNs and on small-scale datasets, specifically CIFAR10 and MNIST. These choices are not convincing enough to demonstrate the method's effectiveness and generalizability across different or more complex datasets. The fact that only subsets of these datasets were used further constrains the results, making it difficult to assess how the proposed method would perform on larger, real-world datasets like ImageNet.
2. The paper fails to include a comprehensive comparative analysis with contemporary methods such as NCE, LEEP, and LogMe. This omission is significant because it does not allow for a clear understanding of how the proposed method stands against the latest advancements in the field, especially those methods that might use similar or more advanced techniques for source selection in transfer learning.
3. The manuscript assumes the use of a fixed source model that will be fine-tuned on the target task, a black-box scenario that diverges from common practice where users typically have access to the pre-trained model. This assumption raises practical concerns: why would users opt to fine-tune a model without access to its original, pre-trained state? This setting contrasts with existing works that usually allow for adjustments to the source model itself. An explanation of the practicality of this assumption and a justification for diverging from the typical settings found in the literature would strengthen the manuscript's relevance and applicability.
4. Measuring the relatedness between the source and target tasks to estimate the transferability has been explored in [1-3]. A detailed discussion comparing these existing methods with the novel approach proposed in the manuscript is necessary. This would highlight the contributions or potential improvements offered by the new metric and provide clarity on its benefits over previous strategies.

[1] Taskonomy: Disentangling Task Transfer Learning. CVPR 2018

[2] OTCE: A Transferability Metric for Cross-Domain Cross-Task Representations, CVPR 2021

[3] Understanding the Transferability of Representations via Task-Relatedness, NeurIPS 2024

**Questions:**

1. In line 372, could you specify what the first and second datasets refer to? Are they different from the third one, i.e. CIFAR10-MNIST?
2. The computational benchmarks in the paper are conducted on a CPU, whereas, in typical practice, model fine-tuning is often performed on a GPU. Given the specific fine-tuning setup described, which involves only updating a few linear layers on top of a fixed pre-trained model, the process is expected to be relatively quick on a GPU. Therefore, it would be valuable to compare the efficiency of the proposed metric against traditional fine-tuning methods when both are executed on a GPU. How does the efficiency of the proposed metric compare to that of standard GPU-based fine-tuning?
3. The evaluation metrics used in this paper differ from those typically seen in related works. Could including Pearson or Kendall coefficients provide a more standardized comparison with existing methods?

---

> ### Author Response · Authors · 2024-11-27
> **Response to weaknesses**
>
> We thank you for your valuable feedback and apologize for the delay in our response as we were trying to add new simulation results to respond to the comments by all the reviewers. We have now added 3 sections in the appendix of the paper (F.G.H) (details announced as a separate response).
>
> > **The experiments are conducted using only DNNs ... datasets like ImageNet.**
>
> Response: Thank you for your feedback.
> 1. The goal of this work was to showcase a proof of concept for the proposed novel quantization-based approach. To achieve this, we utilized smaller datasets, allowing us to produce comprehensive results efficiently within a shorter timeframe.
> 2. We have now added additional simulation results for the Imagenette dataset (a subset of Imagenet) with 64x64 images in Appendix G where both the source and target tasks are binary classification tasks for images in the Imagenette dataset. We have included plots to visualize the source rankings given by our metric and compared them to the true rankings in Section G.1. In G.2, We have results for the fraction of accurate rank predictions for two different target tasks. Finally, in Section G.3, we present the time improvement statistics for using our metric versus training the model to find transfer performance.
>
> > **The paper fails to include a comprehensive comparative analysis with contemporary methods such as NCE, LEEP, and LogMe. ... selection in transfer learning.**
>
> Response:
> 1. Methods like NCE (Tran et al. (2019)), OTCE, and LEEP (Nguyen et al. (2020)) use both the source and target labels to estimate transfer performance. Our approach is different in that it is a very lightweight approach that does not need the source data used to train the source model. This makes our work practically appealing compared to the state-of-the-art. It also means that it is not possible to have a fair comparison of our approach with methods like NCE, OTCE, and LEEP.
> 2. Our setup differs from LogME in that we only have access to the softmax outputs of the pre-trained source model, as we operate under a black-box source setup (justification provided in response to weakness 3). In contrast, LogME explicitly states that it does not utilize the classification head (softmax outputs). Therefore, a direct comparison between our approach and LogME is not feasible.
>
> > **The manuscript assumes the use of a fixed source ... strengthen the manuscript's relevance and applicability.**
>
> Response: Thank you for this question. We envision that with the development of closed-source pre-trained models like ChatGPT, Claude, etc., there will be increase in the availability of private models which can be only accessed through an API (without knowledge of the parameters) and we can only access the probability distribution over the output labels (i.e. softmax output). In such a case, to select the most transferable source, it is practical requirement to evaluate the transfer performance of the source for a given target task. Our proposed transferability measure is aimed to target such scenarios.
>
> > **Measuring the relatedness between the source and target tasks to ... benefits over previous strategies.**
>
> Response:
> 1. In their work in [1], as detailed in Section 3 of their paper, the authors focus on a transferability measure that requires evaluating transfer performance by training a target model. The task taxonomy they propose, represented as a directed hypergraph, is constructed based on these transfer performances. In contrast, our work aims to eliminate the need for training the target model to assess transferability. Hence, the objective of our work is different and it's difficult to have a fair comparison between both.
> 2. As mentioned in the response for weakness 2, existing methods for transferability estimation like OTCE [2], NCE assume that the source data that was used to train the source model is available. Our approach is aimed to study a different setting which does not need the source data, making it lightweight as it only needs the pre-trained source model.
> 3. In their work in [3], the authors assume that the probability distribution of the reference task (source) $P_R(x,y)$ is known. However, this contradicts our setting as we do not have access to the source data and hence would not have this distribution available. Our work only uses the distribution of the softmax output of the source given target label.

---

> > ### Author Response · Authors · 2024-11-27
> > **Response for Questions**
> >
> > > **In line 372, could you specify what the first and second datasets refer to? Are they different from the third one, i.e. CIFAR10-MNIST?**
> >
> > Response: In line 372, we were trying to explain the meaning of the notation used to denote the 3 transfer learning setups. These notations (e.g., MNIST-MNIST) are made up of two datasets where the first and second datasets refer to the dataset used to train the pre-trained source model and the target model respectively. For example, for the third transfer learning setup CIFAR10-MNIST, the source model has been trained to classify images in CIFAR10 while the target model aims to classify images in MNIST. Here, the transfer is being done from representation learned during training to classify CIFAR10 images, to classify images in MNIST.
> >
> > > **The computational benchmarks in the ... that of standard GPU-based fine-tuning?**
> >
> > Response: Thank you for your insightful comment and for highlighting the importance of GPU-based simulations. We chose to conduct the computational benchmarks on a CPU primarily because our approach only requires fine-tuning a few linear layers on top of a frozen source model. Given the limited computational overhead involved in this setup, we found that CPU-based execution was sufficient and efficient for our purposes. However, we are trying to simulate on GPU as well which might take some time to finish. We will report the results once done.
> >
> > > **The evaluation metrics used in this paper differ from those typically seen in related works. Could including Pearson or Kendall coefficients provide a more standardized comparison with existing methods?**
> >
> > Response: We appreciate the suggestion and have worked on it to include results for Spearman, Pearson, and Kendall's Tau correlation in Appendix F under 'Correlation Statistics'. The correlation is between the predicted source rankings (using our metric) and the true rankings (using transfer accuracy) for all 45 source models for different dataset sizes (*tl-frac*) for 3 transfer setups. Just to reiterate, *tl-frac* is the fraction of the entire available dataset used for transfer learning to mimic the standard limited sample transfer setup (*tl-frac=0.01* means 1% of the entire data was used as target data). Overall, we can observe high correlation across transfer setups for different datasets with different dataset sizes indicating good performance of our metric.
> >
> > ---
> > We hope that our detailed comments and the additional data (new appendix sections of the paper) have provided clarity on the questions and concerns raised regarding our paper. We humbly request the reviewer to kindly reconsider their evaluation in light of the updated information, should they find it appropriate.

---

> > > ### Comment · Reviewer_mueD · 2024-11-29
> > >
> > > > **The goal of this work was to showcase a proof of concept for the proposed novel quantization-based approach.**
> > >
> > > While introducing a novel concept is commendable, the method should also demonstrate practical utility. Source model selection is a highly practical and valuable problem in the ML domain. To establish the usefulness of the proposed method, it should be evaluated on larger datasets, across diverse settings, and compared with existing baselines. Even though the authors argue that baselines such as NCE and LEEP are not comparable—a claim I disagree with—the proposed method should still be tested under the same experimental settings. For instance, the experiments in the manuscript are limited to cases where the source and target have the same number of classes.
> > >
> > > > **Methods like NCE (Tran et al., 2019), OTCE, and LEEP (Nguyen et al., 2020) use both the source and target labels to estimate transfer performance. Our approach is different in that it is a very lightweight approach that does not need the source data used to train the source model.**
> > >
> > > This is not entirely accurate. NCE and LEEP do not require access to the source data. These methods only rely on the true labels of the target data and the predictions for the target data in the source label space to compute their metrics.
> > >
> > > > **We envision that with the development of closed-source pre-trained models like ChatGPT, Claude, etc.**
> > >
> > > I agree with the potential relevance of investigating transferability under the assumption of black-box access to generative models, such as LLMs. However, this vision is not reflected in the experiments. The current work addresses classification tasks rather than generation tasks, and it is unclear how the proposed method would perform with generative models like ChatGPT. Additionally, a practical use case for the scenario where only black-box access to a pre-trained classifier is available should be clearly defined. Furthermore, how does the performance of a model fine-tuned on logits from a black-box pre-trained model compare to that of a model trained from scratch on the target dataset? If transfer learning is justified by better performance, the results need to demonstrate this convincingly. As it stands, the setting is not convincing enough.
> > >
> > > > **Given the limited computational overhead involved in this setup, we found that CPU-based execution was sufficient and efficient for our purposes.**
> > >
> > > While CPU-based execution may suffice for the experiments presented, the computational overhead could significantly increase when scaling to large foundation models, such as those mentioned earlier (e.g., ChatGPT). Evaluating the computational complexity on GPUs, especially for more demanding scenarios, is necessary.

---

> > > > ### Author Response · Authors · 2024-12-02
> > > >
> > > > Thank you for your response and follow-up comment.
> > > >
> > > > >**This is not entirely accurate. NCE and LEEP do not require access to the source data ... compute their metrics.**
> > > >
> > > > Response: Initially, we argued that NCE and LEEP were unsuitable baselines for comparison. However, after your comments, we revisited [4], which referred to these methods, and realized we had misinterpreted the meaning, confusing labels of the source domain with source data. We acknowledge this mistake and appreciate your input, which helped us correct our understanding. Nonetheless, our clarification on applicability for [1] [2] [3] (as in your original comment) remains the same.
> > > >
> > > > [4] Transferability Estimation using Bhattacharyya Class Separability
> > > >
> > > > >**To establish the usefulness of the proposed method, it should be evaluated on larger datasets**
> > > >
> > > > Response: We agree that evaluation on larger datasets is always good. However, in the short time that we had for the rebuttal, we tried to include results for the Imagenette dataset (a subset of the Imagenet dataset) in Appendix G.
> > > >
> > > > >**For instance, the experiments in the manuscript are limited to cases where the source and target have the same number of classes.**
> > > >
> > > >  Response: We would like to clarify that our results include scenarios where the source and target tasks have differing numbers of classes. Specifically, in Section 5.2, we consider three cases: (1) a 3-class source to a binary target, (2) a 4-class source to a binary target, and (3) a 4-class source to a 3-class target.
> > > >
> > > > >**I agree with the potential relevance of investigating transferability under the assumption of black-box access to generative models, such as LLMs ... is not convincing enough.**
> > > >
> > > > Response:
> > > > 1. As mentioned earlier, our work focuses on classification tasks. ChatGPT was just an example to motivate the models that can only be accessed through an API and we did not mean to emphasize generative models. For example, Clarifai is a company that provides proprietary solutions to classification tasks (vision and text).
> > > > 2. The black box source model assumption, mathematically speaking, is simply saying that we want to use the distribution over source label space for our transferability metric.
> > > > 3. Existing methods like NCE and LEEP use the softmax output distribution of the pre-trained source model to calculate their respective metrics. These papers do not use the word 'black box' while still using a similar approach to ours. In simple words, the architecture used in our paper has been used in several works.
> > > > 3. Clarifying the last part of the question, we want to emphasize that our work does not focus on justifying the usefulness of transfer learning but rather deals with the question - given several options for source models, which one should we choose for a given target task. Hence, the evaluation of whether training from scratch is better does not align with the core idea of the paper.
> > > >
> > > > >**While CPU-based execution may suffice for the experiments ... especially for more demanding scenarios, is necessary.**
> > > >
> > > > Response: We appreciate the input and will surely incorporate it in the next iteration of our work.
> > > >
> > > > ---
> > > > We hope that our comments have provided clarity on the questions and concerns raised regarding our paper. We humbly request the reviewer to kindly reconsider their evaluation in light of the updated information, should they find it appropriate.

---

> ### Comment · Reviewer_mueD · 2024-12-03
>
> Thank you for your response. I appreciate the clarification provided, but my concerns remain unresolved. Specifically, the unconventional black-box source model setting and a lack of comprehensive comparison with existing methods.
>
> > **"These papers do not use the word 'black box' while still using a similar approach to ours."**
>
> This statement reflects another misunderstanding. When metrics like LEEP or NCE are computed, the source pre-trained model is indeed fixed (like a black box). However, my concern lies with the fine-tuning process. In the referenced papers, the entire model is accessible, allowing for full fine-tuning (tune all parameters) with target data. Conversely, in this manuscript, the black-box setting only adds additional linear layers while freezing the pre-trained model. This limitation can significantly compromise performance.
>
> I strongly recommend that the authors revisit the NCE, LEEP, and LogMe papers. Proposing a good metric for linear fine-tuning—where only one or a few linear layers are updated—is not practically useful, as linear fine-tuning is computationally lightweight and can be done very quickly. In fact, the time required to compute a metric may sometimes exceed the time required for linear fine-tuning itself [3]. Thus, there is little practical value in computing a metric instead of simply assessing the linear fine-tuning performance.
>
> A useful metric should correlate strongly with the performance of **full fine-tuning** (where all model parameters are updated), as full fine-tuning and hyperparameter tuning are computationally expensive. This is also why evaluating the computational time on GPUs is necessary to establish the method’s efficiency.
>
> > **"The evaluation of whether training from scratch is better does not align with the core idea of the paper."**
>
> This critique is closely tied to the unrealistic nature of the black-box setting. The rationale behind training a classifier on a model that cannot be fully accessed is unclear. Unlike black-box generative models, there are numerous publicly available large foundation models, such as CLIP, which already offer strong performance.
>
> Additionally, I searched for Clarifai and found:
>
> > **"Clarifai provides an end-to-end, full-stack enterprise AI platform to build AI faster, leveraging today's modern AI technologies like cutting-edge Large Language Models (LLMs), Generative AI, Retrieval Augmented Generation (RAG), data labeling, inference, and much more."**
>
> This suggests that Clarifai is not a provider of black-box classification models but rather an AI development platform. The manuscript should include a clear and practical use case for the proposed black-box setting to justify its relevance.
>
> > **"Nonetheless, our clarification on applicability for [1] [2] [3] (as in your original comment) remains the same."**
>
> I disagree. The assumption and experimental setup for computing the metrics are the same across the referenced methods, so they are indeed comparable. A thorough comparison with existing methods under same experimental conditions (e.g., datasets, model zoo) would significantly strengthen the paper.
>
> In conclusion, the paper still requires substantial improvements to address these concerns and provide clarity on the proposed setting and method. Additionally, the sluggish rebuttal process has left insufficient time for further discussion. Therefore, I will maintain my initial score.

---

### Official Review · Reviewer_YoTc · 2024-11-03

**Soundness:** 3
**Presentation:** 4
**Contribution:** 2
**Rating:** 5
**Confidence:** 4

**Summary:**

This manuscript introduces a task-similarity metric designed to identify the most transferable source pre-trained model for a given target task. The proposed metric leverages a quantization-based method to evaluate the similarity between a source pre-trained model and a new target dataset without requiring re-training. By using an early stopping technique, the proposed method derives a quantized representation of the source model’s softmax outputs, which could be used to rank and select the best source pre-trained models. Experimental results demonstrate the method’s effectiveness across various classification datasets.

My main concern is how the theoretical framework supports the claim that the proposed quantized representation relates to the generalization of a pre-trained model on a given target task. If the authors can address this concern properly during rebuttal, I am inclined to increase my evaluation score significantly; otherwise, I may consider lowering it further.

############################### Post Rebuttal ###############################

None author response found. I will keep my score.

**Strengths:**

1. Overall, the paper is well-written with clear motivation. Each of the proposed component and its theoretical insights are clearly presented.

2. This manuscript provides a practical strategy for selecting the most transferable source models, allowing for efficient pre-selection before engaging in the computationally intensive transfer to target tasks.

3. The proposed method introduces a task-similarity metric based on quantization-level optimization, which is novel and interesting

**Weaknesses:**

1. Based on my understanding, the quantization level is an important aspect of the proposed method. However, there is no experiment showing how changes in the quantization level affect target task performance. I suggest that the authors include a detailed ablation study on the quantization level.

2. As mentioned by the authors, the proposed method relies on early stopping during source pre-training. Thus, repeated experiments with different random seeds for network initialization are necessary, as early stopping can be influenced by these random seeds. I suggest the authors conduct repeated experiments and include the standard deviation of the results in their figures to reflect this variability.

3. The experiments are conducted on small and synthetic image datasets, and early stopping might be influenced by task type and data modality. I suggest that the authors conduct more experiments on various data modalities (e.g., time-series, language) and tasks (e.g., regression) to better validate the effectiveness of the proposed source model selection strategy.

4. While the manuscript focuses on selecting the best source model, it is unclear how well it can handle negative transfer, where an unsuitable source pre-trained model could harm target task performance. I would suggest some visualizations demonstrating how the proposed strategy can handle negative transfer.

5. Certain theoretical assumptions, such as how the target generalization relies on the early stopping and the quantization strategy, are not well-justified or well-connected in the context of transfer learning. At least, I would expect a discussion on how an upper bound of the target classification error might be connected to early stopping of the source pre-trained models with different underlying tasks.

6. Early stopping is a well-studied method for better understanding a model’s generalization in classification tasks. However, the experiments lack baseline methods for comparison. Without comparisons to existing methods with similar objectives, it is difficult to evaluate the practical value of the proposed method. I would suggest the authors to conduct an in-depth literature review on early stopping.

**Questions:**

1. How sensitive is the method to changes in the quantization level, and are there any scenarios where finding an optimal  $q^{\ast}$  might not be feasible?

2. Can the proposed method be applied to tasks beyond classification, such as regression?

---

> ### Comment · Reviewer_YoTc · 2024-11-26
> **Post Rebuttal**
>
> None author response found. I will keep my score.

---

> > ### Author Response · Authors · 2024-11-30
> > **Request to read the author responses**
> >
> > Dear Reviewer,
> >
> > We kindly request you to review our responses to your comments at your convenience. Your feedback is invaluable, and we would greatly appreciate your thoughts on our clarifications and revisions. We apologize for the delay in our response as it took a while to add new simulation results to respond to the comments by all the reviewers. We have now added 3 sections in the paper's appendix (F.G.H) (details announced as a separate response).
> >
> > Thank you for your time and consideration.

---

> > > ### Comment · Reviewer_YoTc · 2024-12-01
> > > **Reply from Reviewer YoTc**
> > >
> > > I believe the authors submitted their response right at the rebuttal deadline. Anyway, here are my follow-up comments on their response:
> > >
> > > > As mentioned by the authors, the proposed method relies on early stopping during source pre-training. ... their figures to reflect this variability.
> > >
> > > I was asking how different stopping points during source pre-training (early stopping) influence target adaptation. Should the ablation study focus on how the number of epochs in source pre-training affects target classification rather than the impact of the random seed?

---

> > > > ### Author Response · Authors · 2024-12-01
> > > >
> > > > Thank you for your response and follow-up comment. In our updated version, in Appendix H, we addressed the specific request as we understood it from your initial comments regarding repeated experiments with different random seeds for network initialization.
> > > >
> > > > In response to your follow-up comment, we would clarify a few things.
> > > > 1. The training of the source model can be broadly done under two settings - fixed epoch and early stopping, where the early stopping has a condition on the validation loss/accuracy (ex - stop if validation loss does not decrease for 10 epochs). This is a dynamic setting in which the training stop epoch cannot be predetermined. Different random seeds to initialize the same source model can result in early stopping at a different epoch, hence a different source model and a different transfer accuracy for the same data.
> > > > 2. To study the effect of the number of epochs in source pre-training, we need to perform a fixed-epoch training which is different than the generally used early stopping method (also used in the paper).
> > > > 3. Our aim was never to focus on training the best source model for a source task. Instead, we focus on **given a source model**, can we measure its transferability to a target task? Hence, if we stop training a source model before its optimal early stopping epoch, we would have a suboptimally trained source model which might not reflect the true transfer performance for a given target. However, this bad transfer will **still be reflected in our metric**.
> > > > 4. In simple words, there cannot be a meaningful relation between the **number of epochs for source training** and **transfer performance** since for the same number of epochs for different source models, the **quality** of the source model varies and hence the transferability too.
> > > >
> > > > We hope this provides some clarification of your question. Since the discussion period has been extended, we look forward to engaging in meaningful discussion.

---

> ### Author Response · Authors · 2024-11-27
> **Response to weaknesses**
>
> We thank you for your valuable feedback and apologize for the delay in our response as it took a while to add new simulation results to respond to the comments by all the reviewers. We have now added 3 sections in the appendix of the paper (F.G.H) (details announced as a separate response).
>
> > **Based on my understanding, the quantization level ... study on the quantization level.**
>
> Response: We believe that there might be some misunderstanding regarding the quantization talked in the paper. The quantization of the softmax values of the pre-trained source model is only used to get our transferability metric (Equation 5) and has no significance in determining the target task performance. The target task performance is defined as the test accuracy when the target model (illustrated in Figure 1), is trained keeping the pretrained part frozen.
>
> > **As mentioned by the authors, the proposed method relies on early stopping during source pre-training. ... their figures to reflect this variability.**
>
> Response: Thank you for the comment. We have added Appendix H - 'Variation with different random seeds for pre-trained source' with the simulation results using 5 different random seeds for source model initialization before training. Specifically, in Fig H.1, we show the variation of $A^{tr}$ and $A^{val}$ (train and validation accuracy) following a similar structure for different random seeds for MNIST-MNIST and CIFAR10-CIFAR10 transfer setups. In Fig H.2, we present the fraction of the correct rank for the CIFAR10-CIFAR10 transfer setup with mean and standard deviation across random seeds. Owing to the limited time frame of the rebuttal, we used these two transfer setups.
>
> > **The experiments are conducted ... model selection strategy.**
>
> Response: Our problem statement for transferability measure primarily focused on classification tasks. Hence, our approach cannot be extended directly to time-series and regression tasks. However, we acknowledge that it is a good point and we will try to work to incorporate it in our next work.
>
> > **While the manuscript focuses on selecting the best source model, ... handle negative transfer.**
>
> Response:
> 1. A source model in the given set of sources can have varying degrees of transferability for a given target, going from low to high transfer accuracy. The proposed metric is a quick measure of this transferability. When the metric is run for a source that is unrelated to the target, the metric gives a value that ranks it amongst the worst performing in the set. Here, the unrelated sources can be considered as contributing to negative transfer (note that negative transfer is a spectrum as there is no well-defined threshold to be called negative transfer).
> 2. To show that our proposed metric handles negative transfer cases well, refer to Figure 5, where the metric values (blue) and the transfer learning accuracy (red) are represented for different source models for 3-class source to binary target tasks for MNIST-MNIST transfer setup. We can see that source tasks with the worst transfer performance (~60% accuracy) like source to classify digits (0,3,8), (0,1,5), (0,5,8), etc., have their corresponding metric values very low indicating bad transfer performance.
>
> > **Certain theoretical assumptions, such as how the ... with different underlying tasks.**
>
> Response: This is an initial work for this problem statement where we wish to demonstrate a novel quantization-based approach. Studying the theoretical aspects of the relationship of the early stopping point with the generalization of the target model is our aim for future work. However, intuitively, there is a relation between the accuracy of the trained target model and the metric value (calculated without training) since in both cases, we try to maximize the training performance (accuracy/loss) while monitoring the validation behavior. We stop when the validation performance degrades, which is common in both setups, the difference being we monitor validation performance over time in traditional training but over quantization levels $q$ in metric calculation.
>
> > **Early stopping is a well-studied method for ... literature review on early stopping.**
>
> Response: We wish to clarify that we do not aim to propose a new early-stopping method. We use a very commonly used early stopping method i.e. stop when validation performance (accuracy in our case) starts decreasing. Our approach uses this concept of early stopping to select an optimal quantization level $q^*$ (unlike the epoch for when to stop in traditional early stopping). The novelty is to use this well-known concept in a different problem i.e. problem to choose an optimal quantization level. Therefore, a direct comparison between traditional early stopping baselines and our method is not applicable.

---

> ### Author Response · Authors · 2024-11-27
> **Response to questions**
>
> > **How sensitive is the method to changes ... might not be feasible?**
>
> Response: Thank you for your question.
> 1. Interpreting sensitivity as the change in $A^{val}$ as quantization level $q$ changes, we can say that it depends on several factors like the quality of the pre-trained source (related or unrelated to target) and the target data. For source models with low transfer performance ($\sim$50% accuracy), the curve for $A^{val}$ vs $q$ (similar to the one illustrated in Fig 4) starts around 50% and remains around the same as $q$ changes with small zig-zag-like variations. However, for good source models (transfer accuracy of $\sim$80% or more), we can see there is a large change in $A^{val}$ with a small change in $q$ at first (as in Fig 4) and after the optimal $q^*$, the decrease in $A^{val}$ is rather slow. Hence, the sensitivity varies.
> 2. By definition of our method (Eq 5), it is always possible to find an optimal $q^*$ as we can always run both a brute force search and a ternary search (Algorithm 1) on the fixed search space for $q$.
>
> > **Can the proposed method be applied to tasks beyond classification, such as regression?**
>
> Response: This paper addresses the problem setup for transfer learning in classification tasks and was not developed with regression tasks in mind. Therefore, applying our method directly to regression tasks would be challenging.
>
> ---
> We hope that our detailed comments and the additional data (new appendix sections of the paper) have provided clarity on the questions and concerns raised regarding our paper. We humbly request the reviewer to kindly reconsider their evaluation in light of the updated information, should they find it appropriate.

---

### Author Response · Authors · 2024-11-27
**Additional results in the appendix**

Dear reviewers,

We thank you all for your valuable and insightful feedback. Based on your comments, we have added new simulation results in the paper (described below) in the appendix section.
1. **Appendix F - Correlation Statistics** - in addition to evaluating the source rankings provided by the metric vs true rankings using the fraction of correct rank predictions (in Section 5), we have added the correlation statistics using Spearman, Pearson and Kendall's Tau correlations.
2. **Appendix G - Experiments with Imagenette Dataset** - we ran addition simulations for the Imagenette dataset (subset of Imagenet dataset) for 64x64 images and presented the results involving - visualization of the predicted vs true rankings, fraction of correct rank predictions, and time improvement statistics.
3. **Appendix H - Variation with different random seed for pre-trained source** - as per the comment of one of the other reviewers, since the method depends on the pre-trained source model, we tried to run a few simulations with 5 different random seeds to initialize the pre-trained source model parameters before training. The results are presented in this section indicating that our results are not dependent on a particular network initialization.

We hope this helps clarify some of the concerns and strengthen the quality of our work. We humbly request the reviewers to kindly reconsider their evaluation in light of the updated information, should they find it appropriate.

---

### Meta-Review · Area_Chair_k7Cz · 2024-12-15

**Metareview:**

This paper proposes BeST, a quantization-based metric for selecting the most transferable pre-trained source models in transfer learning. The method leverages early stopping and quantization to estimate transferability without full training, aiming to reduce computational costs. Experiments were conducted on datasets like MNIST and CIFAR10.

**Strengths:**

The metric is innovative, computationally efficient, and theoretically sound.

It shows strong correlations with transfer performance and achieves significant time savings in source selection.

**Weaknesses:**

Experiments are limited to small datasets and lack evaluation on larger or cross-domain tasks.

The absence of comparisons with existing methods like LogME, LEEP, and NCE weakens its impact.

Practical use cases for the black-box source model setting are not convincingly demonstrated.

**Recommendation:**

Reject.

While novel, the paper requires broader evaluation, comparative analysis, and clearer articulation of its practical relevance to make a stronger contribution.

**Additional Comments On Reviewer Discussion:**

Unfortunately, Reviewer g5gx did not respond to the authors' rebuttal. But, from the other two reviewers, YoTc and mueD, it is seen that they are still concerned with insufficient experimental analysis and clarity of the proposed setting and method etc.

---

### Decision · Program_Chairs · 2025-01-22

Reject